# Pulsed laser induced plasma and thermal effects on molybdenum carbide for dry reforming of methane

Yue Li [1,5], Xingwu Liu [2,5], Tong Wu[1], Xiangzhou Zhang[1], Hecheng Han[3], Xiaoyu Liu[4], Yuke Chen[1], Zhenfei Tang[1], Zhen Liu[1], Yuhai Zhang [1], Hong Liu[1,4], Lili Zhao [1] ✉, Ding Ma [2] ✉ & Weijia Zhou [1] ✉

Dry reforming of methane (DRM) is a highly endothermic process, with its development hindered by the harsh thermocatalytic conditions required. We propose an innovative DRM approach utilizing a 16 W pulsed laser in combination with a cost-effective $Mo_2C$ catalyst, enabling DRM under milder conditions. The pulsed laser serves a dual function by inducing localized high temperatures and generating $^*CH$ plasma on the $Mo_2C$ surface. This activates $CH_4$ and $CO_2$, significantly accelerating the DRM reaction. Notably, the laser directly generates $^*CH$ plasma from $CH_4$ through thermionic emission and cascade ionization, bypassing the traditional step-by-step dehydrogenation process and eliminating the rate-limiting step of methane cracking. This method maintains a carbon-oxygen balanced environment, thus preventing the deactivation of the $Mo_2C$ catalyst due to $CO_2$ oxidation. The laser-catalytic DRM achieves high yields of $H_2$ (14300.8 mmol $h^{-1}$ $g^{-1}$) and CO (14949.9 mmol $h^{-1}$ $g^{-1}$) with satisfactory energy efficiency (0.98 mmol $kJ^{-1}$), providing a promising alternative for high-energy-consuming catalytic systems.

The extensive emission of greenhouse gases, specifically methane ($CH_4$) and carbon dioxide ($CO_2$), is a primary cause of global warming[1,2]. Catalytic dry reforming of methane (DRM) offers an environmentally friendly and viable route for large-scale greenhouse gases utilization ($CO_2 + CH_4 \rightarrow 2CO + 2H_2$, $\Delta H_{298K} = +247 kJmol^{-1}$), converting two gases into valuable chemical feedstock[3,4]. Unfortunately, DRM is an endothermic reaction, requiring high temperatures (700–1000 °C) to overcome unfavorable dynamics and thermodynamics[5]. At present, thermocatalysis driven by external heating was limited by high temperature and high pressure, resulting in high energy consumption. Photothermal catalysis occurred at a relatively low temperature, but the catalytic efficiency possessed major room for improvement[6]. Furthermore, due to the high-

temperature reaction, coke deposition and concomitant catalyst deactivation during DRM are inevitable problems[7]. Consequently, it is imperative to explore novel catalytic modalities that combine mild reaction conditions with high catalytic performance.

The thriving exploration of catalysts for DRM has been witnessed in the past decades. Especially, the Group VIII metals, whether precious metals (Rh, Ru, Pt)[3,8–10] or non-precious metals (Ni, Co, Fe)[11–13], have been proven to possess catalytic activity in DRM reactions. Transition metal carbides, such as molybdenum carbide ($Mo_2C$) and tungsten carbide (WC)[14,15], exhibited remarkable catalytic activity in methane aromatization reaction[16,17]. However, under thermal catalytic reaction condition, $Mo_2C$ possesses limited methane cracking ability in DRM reaction[18], which leads to over-oxidation and eventually the

[1]Institute for Advanced Interdisciplinary Research (iAIR), School of Chemistry and Chemical Engineering, University of Jinan, Jinan, China. [2]Beijing National Laboratory for Molecular Sciences, College of Chemistry and Molecular Engineering, Peking University, Beijing, China. [3]Shandong Technology Center of Nanodevices and Integration, School of Integrated Circuit, Shandong University, Jinan, China. [4]State Key Laboratory of Crystal Materials, Shandong University, Jinan, China. [5]These authors contributed equally: Yue Li, Xingwu Liu. ✉e-mail: ifc_zhaoll@ujn.edu.cn; dma@pku.edu.cn; ifc_zhouwj@ujn.edu.cn

deactivation of the catalyst in carbon-deficient environments[19–21]. Hence, improving the methane cracking capacity is of great significance to the Mo$_2$C catalyst.

Pulsed laser is a type of high-energy density light produced by the process of stimulated emission and light amplification[22]. Owing to its narrow pulse width and high-energy density, the interaction between pulsed laser and materials can give rise to some fascinating effects[23,24]. The local thermal effect produced by pulsed laser can trigger photo-thermochemical reactions[25–27]. More importantly, high-energy electrons generated by an infocus pulsed laser can cause the breakdown of gas molecules, resulting the formation of highly active plasma from the gas reactants[28–30]. Studies have confirmed that plasma enabled a thermodynamic-limited reaction to occur with a fast reaction rate at low temperatures[31,32]. It is worth noting that, unlike pulsed laser, the continuous wave (CW) laser usually produced a high temperature instead of plasma due to the efficient photothermal conversion via non-radiative relaxation[33]. Consequently, it is rational to speculate that pulsed laser can be applied in DRM as both heat and plasma sources, which represent an efficient supplementary to the deficiency of Mo$_2$C.

Here, we reported that by using the pulsed laser to drive the DRM reaction at relatively mild reaction condition. Record-high activities of H$_2$ (14300.8 mmol·h$^{-1}$·g$^{-1}$) and CO (14949.9 mmol·h$^{-1}$·g$^{-1}$) were reached over a simple Mo$_2$C catalyst by this pulsed laser-driven DRM reaction. The thermal and plasma effect on Mo$_2$C catalyst induced by a 16 W

pulsed laser were identified to be critical for the pulsed laser-driven DRM reaction. More importantly, the pulsed laser-induced CH plasma ($^\cdot$CH) avoided the step-by-step dehydrogenation of CH$_4$, which boosted methane cracking on the surface of the catalyst, established a surface carbon-oxygen equilibrium and protected Mo$_2$C from over-oxidation, which is essential for the high activity and stability of the catalyst. This finding holds significant implications for expanding the research concepts within photothermal catalytic systems.

## Results

### Characterization of Mo$_2$C/BaSO$_4$ tablet

Mo$_2$C nanosheets were synthesized with a rough surface, as depicted in Fig. 1a. The crystal phase of Mo$_2$C was confirmed by X-ray diffraction analysis (Fig. 1b), which exhibited distinct peaks of β-Mo$_2$C. The HRTEM image (Fig. 1c) showed lattice spacings of 0.228 nm (assigned to (101) planes) and 0.237 nm (assigned to (002) planes) for β-Mo$_2$C, confirming the successful synthesis of β-Mo$_2$C. Before the laser-catalytic DRM, the Mo$_2$C/BaSO$_4$ tablet was pressed by Mo$_2$C and BaSO$_4$ powders (Fig. 1d). The catalytically inert BaSO$_4$ was selected as the substrate and didn't affect the catalytic activity of Mo$_2$C (Supplementary Fig. 1, 2). The top Mo$_2$C layer exhibited a thickness of ~36 μm, as evident from the cross-sectional SEM image in Fig. 1e. The corresponding EDS mapping of Mo, C, Ba, and O elements in Fig. 1f also confirmed successful construction of razor-thin Mo$_2$C layer on BaSO$_4$ substrate.

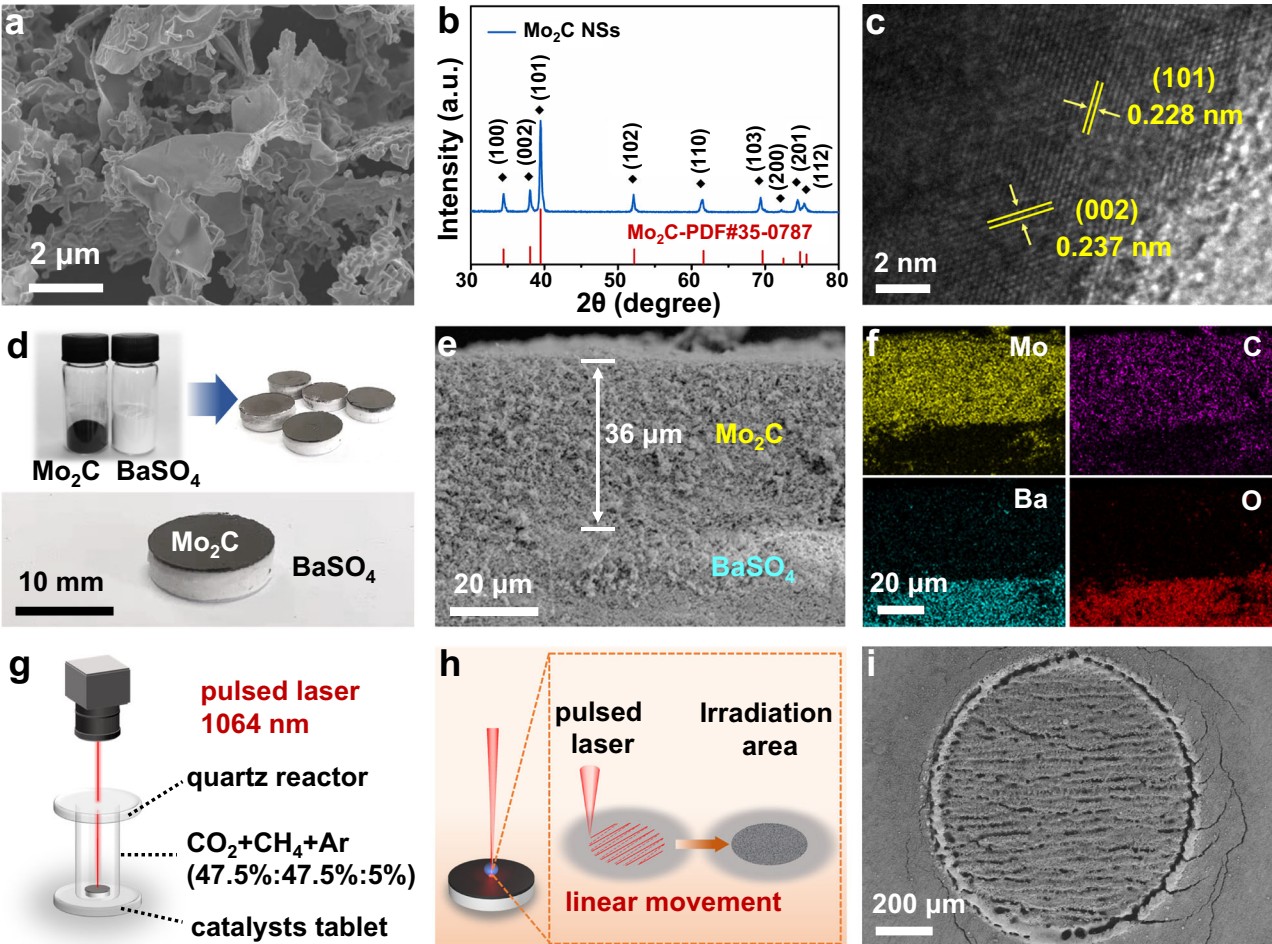

**Fig. 1 | Characterization of Mo$_2$C/BaSO$_4$ tablet and laser irradiation effects.**
**a** SEM image, **b** XRD pattern, and **c** HRTEM image of Mo$_2$C powder. **d** Photograph of tablets pressed by Mo$_2$C and BaSO$_4$ powders. **e** SEM image and **f** corresponding EDS element mapping of the cross-section of Mo$_2$C/BaSO$_4$ tablet. **g** Schematic diagram of laser-catalytic DRM reaction via Mo$_2$C/BaSO$_4$ tablet. **h** Schematic diagram of the

Mo$_2$C/BaSO$_4$ tablet illuminated by pulsed laser and **i** SEM images of the circular spot of Mo$_2$C/BaSO$_4$ tablet irradiated by pulsed laser (the laser irradiates a specific area in a line scanning mode, taking a circular area with a diameter of 0.94 mm as an example).

The experimental equipment of laser-catalytic DRM is shown in Fig. 1g and Supplementary Fig. 3a. During the laser-catalytic DRM, the obtained Mo$_2$C/BaSO$_4$ tablet was placed at the bottom of the quartz reactor. A 16 W fiber optic laser with a laser wavelength of 1064 nm and a pulse duration of 100 ns was used to irradiate the Mo$_2$C/BaSO$_4$ tablet. As depicted in Fig. 1h, laser irradiation in a small area (demonstrated by a circular area of 0.94 mm diameter) in a line sweep was performed. The pulsed laser was concentrated on a tiny point on the surface of Mo$_2$C/BaSO$_4$ tablet. SEM image in Fig. 1i exhibited that the irradiated tiny area formed a regular overheating morphology due to the localized high temperature generated by the laser. As a consequence, it was reasonably speculated that the effective mass of the catalyst in the laser-catalytic DRM process was only in the region treated by the laser spot.

## Laser-catalytic DRM performance

To gain more insight, the laser-catalytic DRM performances of Mo$_2$C/BaSO$_4$ tablets with different tablet areas of 12.57, 7.07, 3.14, and 0.79 mm$^2$ were discussed (Fig. 2a and b, Supplementary Fig. 4 and Fig. 5, Supplementary Table 1-3). The product rate of H$_2$ (6.864 ~ 7.330 mmol h$^{-1}$) and CO (7.176 ~ 7.611 mmol h$^{-1}$) did not change significantly when the tablet area was not less than 3.14 mm$^2$. Thus, it is indicated that laser-catalysis occurs in a tiny area, which is

consistent with the SEM image of the laser-irradiated area (Fig. 1i). However, when the tablet area was reduced to 0.79 mm$^2$, the thin Mo$_2$C layer was easily stripped by the laser resulting in a decrease in the catalytic reaction rate after 20 mins irradiation (Fig. 2c). Consequently, it is deduced that an overly small catalyst area is not beneficial for maintaining catalyst stability, and a tablet area of 3.14 mm$^2$ appears to be optimal. Additionally, we discussed the laser-catalytic DRM performances of Mo$_2$C/BaSO$_4$ tablets with different Mo$_2$C thicknesses (36, 54, 110, 220, and 315 μm). Remarkably, the product rates of H$_2$ (6.998 ~ 6.276 mmol h$^{-1}$) and CO (7.176 ~ 6.201 mmol h$^{-1}$) did not exhibit significant changes as the Mo$_2$C thickness increased. This further underscores that the actual catalytic dose interacting with the laser remains very small, regardless of whether it operates at the area or depth level.

To elucidate the advantages of laser-catalytic DRM, the product yields of thermocatalytic DRM, as well as photothermal catalytic DRM driven by Xenon lamp, were investigated (Fig. 2d). Specifically, the optimal reactive activity of thermocatalytic DRM at 900 °C with yields of H$_2$ (29.4 mmol h$^{-1}$ g$^{-1}$) and CO (97.4 mmol h$^{-1}$ g$^{-1}$) was achieved (Supplementary Fig. 7), which was in good accordance with previous studies[18,34]. The performance of Xenon-lamp-driven photothermal catalytic DRM was also unsatisfying, only 0.3 mmol h$^{-1}$ g$^{-1}$ (yield of H$_2$) and 0.8 mmol h$^{-1}$ g$^{-1}$ (yield of CO) were detected (Fig. 2d), which was

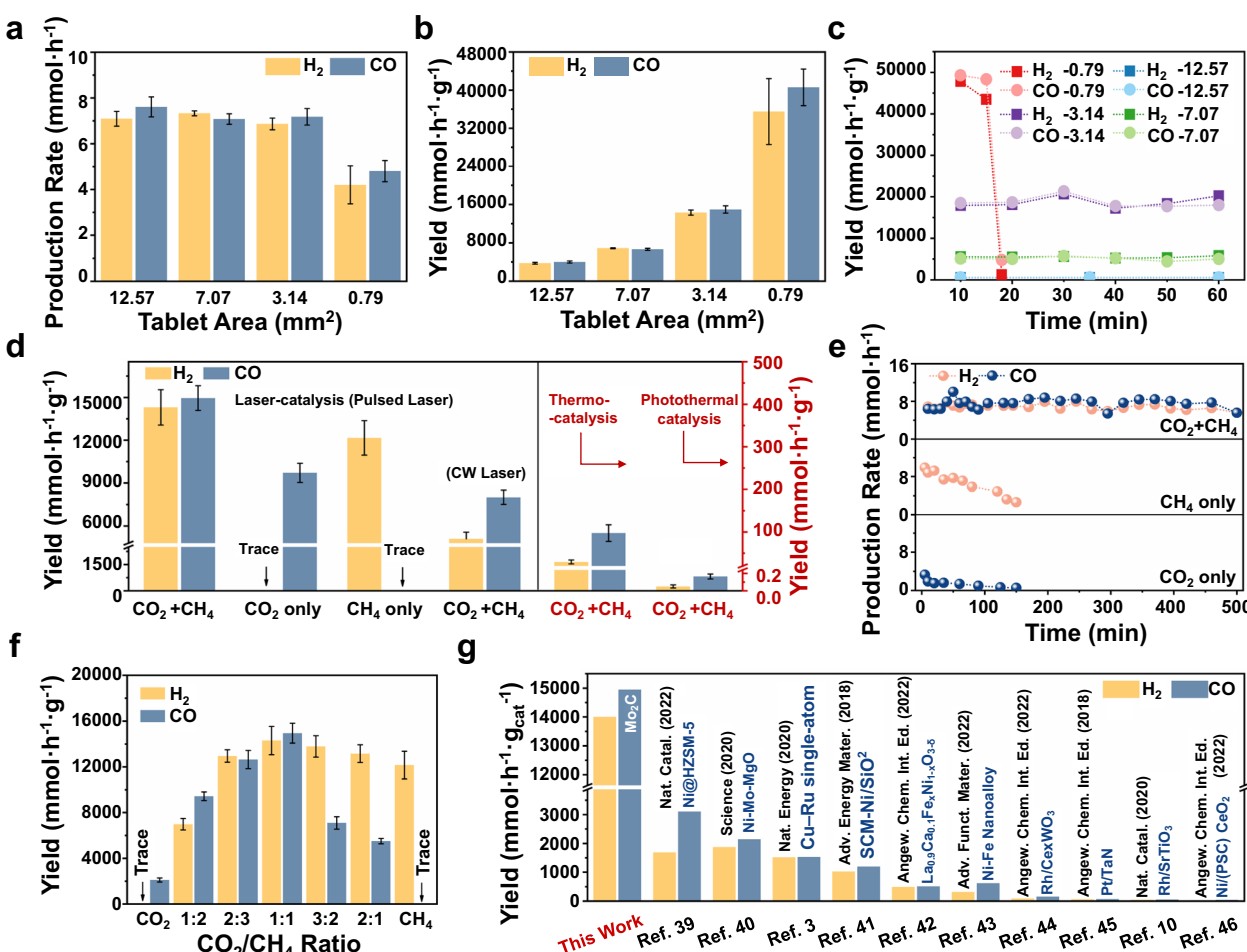

**Fig. 2 | Performance evaluation of laser-catalytic DRM. a, b** Laser-catalytic DRM performances and **c** catalytic stability of Mo$_2$C/BaSO$_4$ tablets with different areas (laser-catalysis in closed system, laser output power: 16 W, infocus mode, CO$_2$:CH$_4$:Ar = 47.5%:47.5%:5%). Error bars represent standard deviation. **d** DRM performance in laser-catalysis, thermocatalysis (temperature: 900 °C), and Xenon lamp-driven photothermal catalysis (optical power density: 3 W cm$^{-2}$). Error bars

represent standard deviation. **e** Catalytic stability under CO$_2$ (CO$_2$:Ar = 95%:5%), CH$_4$ (CH$_4$:Ar = 95%:5%), and CO$_2$ + CH$_4$ (CO$_2$:CH$_4$:Ar = 47.5%:47.5%:5%) atmosphere. **f** Laser-catalytic DRM performances of Mo$_2$C/BaSO$_4$ tablet in different proportions of CO$_2$ and CH$_4$. Error bars represent standard deviation. **g** The comparison of product yield for laser-catalytic DRM in this work and previously reported results.

attributed to the low photothermal temperature (maximum 361 °C at 3 W cm$^{-2}$ in Supplementary Fig. 8) and poor methane activation capability of low-energy photons. The yields of the continuous wave (CW) laser-driven DRM were substantially lower in comparison to those of the pulsed laser, yielding only 5099.8 mmol h$^{-1}$ g$^{-1}$ (for H$_2$) and 8000.6 mmol h$^{-1}$ g$^{-1}$ (for CO). Remarkably, the ultrahigh yields of H$_2$ of 14300.8 mmol h$^{-1}$ g$^{-1}$ and CO of 14949.9 mmol h$^{-1}$ g$^{-1}$ via pulsed laser-catalytic DRM without the assistance of external heating source were obtained (Fig. 2d and Supplementary Fig. 9), which were about 486 times (for H$_2$) and 153 times (for CO) higher than those of thermocatalysis and about 56975 times (for H$_2$) and 18143 times (for CO) higher than those of Xenon lamp-driven photothermal catalysis. Even without comparing the mass of the catalysts, the production rate (6.864 mmol h$^{-1}$ for H$_2$ and 7.176 mmol h$^{-1}$ for CO) of the laser-catalysis system with less catalyst (0.48 mg Mo$_2$C) was still better than that (1.471 mmol h$^{-1}$ for H$_2$ and 4.870 mmol h$^{-1}$ for CO) of the thermocatalysis system (50 mg Mo$_2$C), suggesting the superior intrinsic catalytic activity of laser-driven DRM (Supplementary Fig. 10). Under the same small amount (0.48 mg), the production rate of H$_2$ (0.009 mmol h$^{-1}$) and CO (0.037 mmol h$^{-1}$) and yields of H$_2$ (17.7 mmol h$^{-1}$ g$^{-1}$) and CO (77.2 mmol h$^{-1}$ g$^{-1}$) in thermocatalysis were also much lower than those of laser catalysis (yields of H$_2$ of 14300.8 mmol h$^{-1}$ g$^{-1}$ and CO of 14949.9 mmol h$^{-1}$ g$^{-1}$) (Supplementary Fig. 11). This confirmed that thermocatalysis could not achieve high catalytic activity under a small amount of catalysts, only laser catalysis can achieve high DRM activity under such a small amount of catalysts, which embodied the advantage of laser catalysis.

To illuminate the pulsed laser-catalytic DRM reaction of the Mo$_2$C/BaSO$_4$ tablet, the products under different reaction atmospheres were detected at the same pulsed laser irradiation condition. As shown in Fig. 2d, under CO$_2$ atmosphere, only CO with a low yield of 9707.9 mmol h$^{-1}$ g$^{-1}$ was detected. Under CH$_4$ atmosphere, only H$_2$ was detected, and with the high H$_2$ yield of 12161.3 mmol h$^{-1}$ g$^{-1}$, which verified the strong methane cracking capacity of pulsed laser with Mo$_2$C as laser absorber and catalyst. Whether in single CO$_2$ or CH$_4$ atmosphere, the catalytic reactions were unstable, with significant attenuation observed, from 9.860 to 2.608 mmol h$^{-1}$ for H$_2$ and from 3.288 to 0.550 mmol h$^{-1}$ for CO. In contrast, under CO$_2$/CH$_4$ (1:1) atmosphere, the laser-catalytic DRM with the Mo$_2$C/BaSO$_4$ catalyst exhibited relatively satisfactory stability for 500 mins (Fig. 2e), even more than 50 hours (Supplementary Fig. 12). The XRD patterns (Supplementary Fig. 13) and HRTEM images (Supplementary Fig. 14) revealed that Mo$_2$C phase remained intact under CO$_2$: CH$_4$ feed ratio of 1:1 atmosphere. Although isotopic labelling experiments confirmed that the carbon in Mo$_2$C may participate in the carbon cycle of DRM reaction (Supplementary Fig. 15), the stable existence of the final Mo$_2$C phase and the long-term catalytic stability indicated the establishment of the C-O equilibrium reaction.

In addition, the DRM yields of Mo$_2$C/BaSO$_4$ tablet in varying CO$_2$/CH$_4$ ratios from 2:1 to 1:2 also confirmed the strong methane cracking capacity of pulsed laser. Increasing the CH$_4$ amount during the laser-catalytic DRM resulted in heightened H$_2$ production, as shown in Fig. 2f. In contrast, even if the proportion of CH$_4$ in the reactants was increased, the product of thermocatalytic DRM remained predominantly CO (Supplementary Fig. 16–18), demonstrating the weak methane cracking capacity of Mo$_2$C. Raman and XPS results of Mo$_2$C after laser-catalytic and thermocatalytic DRM also confirmed the above conclusion. The C consumption (Supplementary Fig. 19a) and pronounced oxidation of Mo$_2$C (Supplementary Fig. 19b), arising from its weak methane cracking capacity during thermocatalytic DRM, led to poor catalytic stability. In previously reported results, loading methane activation sites on Mo$_2$C is a common catalyst design strategy, such as creating Metal-Mo$_x$C dual-site catalysts (such as Ni-Mo$_2$C, Co-Mo$_2$C, etc.)[35–38]. Herein, with the aid of the pulsed laser, the cracking of CH$_4$ was significantly boosted and C-O equilibrium on Mo$_2$C surface

was established, yielding excellent DRM activity and stability with pure Mo$_2$C as the catalyst. The laser-catalytic DRM reaction of Mo$_2$C/BaSO$_4$ tablet possessed the high activity of 14300.8 mmol h$^{-1}$ g$^{-1}$ (yield of H$_2$), 14949.9 mmol h$^{-1}$ g$^{-1}$ (yield of CO) and stability (No significant decay over 50 h), which were superior to recently reported DRM results, as summarized in Fig. 2g and Supplementary Table 4[3,10,39–47]. The mass activity of the laser-catalytic DRM without external heating using Mo$_2$C as a catalyst was the highest value up to now in the fields of thermocatalysis and photothermal catalysis.

## Laser-induced localized high temperature

The UV-Vis-NIR absorption spectra depicted in Fig. 3a for both Mo$_2$C and BaSO$_4$ revealed that Mo$_2$C possessed an ideal capacity for laser-induced heat generation due to its effective absorption at 1064 nm wavelength. Due to the negligible absorption ability to laser, BaSO$_4$ could only be heated up to 60 °C under laser irradiation (Supplementary Fig. 20), which caused negligible DRM performance. The radial temperature distribution on the Mo$_2$C/BaSO$_4$ tablet was illustrated in Fig. 3b during laser irradiation, revealing a gradual decline from the central focal point outward. The laser's focused temperature reached 772 °C, which thermodynamically sufficed to propel the DRM reaction. The local high temperature generated by laser irradiation on Mo$_2$C/BaSO$_4$ tablet is one of the prerequisites for DRM reaction. The incident laser was absorbed within the skin depth of the Mo$_2$C surface, instantaneously transforming into heat within sub-nanosecond intervals. Of course, the local temperature of the laser is currently difficult to measure accurately, which is a problem for the industry. The current temperature test is the measurement of the average temperature within a certain region, the actual local temperature value may be different from the measured temperature, but we are unified test conditions and test equipment, to ensure that the temperature trend is accurate.

The effect of laser power on both temperature and product yields in laser-catalytic DRM was explored, as presented in Fig. 3c, d. With the pulsed laser power ascending from 4 W to 8 W, 12 W, and 16 W, the temperature on the Mo$_2$C/BaSO$_4$ tablet escalated from 542 °C to 550 °C, 619 °C, and 772 °C, respectively. The according laser-catalytic DRM reaction performance of H$_2$ and CO yields was improved from 0/45.2 mmol h$^{-1}$ g$^{-1}$ to 132.0/755.6 mmol h$^{-1}$ g$^{-1}$, 645.5/2748.7 mmol h$^{-1}$ g$^{-1}$ and 14300.8/14949.9 mmol h$^{-1}$ g$^{-1}$. In addition, no noticeable difference in temperature was detected under 16 W (772 °C) and 20 W (784 °C). However, the yield of H$_2$ was obviously improved from that of 16 W (14300.8 mmol h$^{-1}$ g$^{-1}$) to that of 20 W (19736.0 mmol h$^{-1}$ g$^{-1}$), implying the thermal effect was not the sole driver of the enhanced performance of laser-catalytic DRM. In particular, the H$_2$/CO molar ratio was significantly increased with increased laser power, suggesting that pulsed laser boosted the CH$_4$ cracking capacity. Furthermore, the effect of different defocusing amounts (Supplementary Fig. 31 and Supplementary Table 5) on laser-catalytic DRM performance also confirmed the above speculation. While the temperatures remained relatively consistent in both under focus (defocusing amount = 20, 15, 10, 5 mm) and in focus modes (defocusing amount = 0 mm), ranging from 746 - 793 °C (Fig. 3e and Supplementary Table 6), the laser-catalytic DRM performances exhibited significant discrepancies, as shown in Fig. 3f. In the in focus mode, the H$_2$/CO yields of 14300.8/14949.9 mmol h$^{-1}$ g$^{-1}$ were significantly higher than those in the under focus modes (3212.8/4953.9, 4681.0/7654.8, 6813.9/10094.0 and 6160.8/9389.7 mmol h$^{-1}$ g$^{-1}$). These results confirmed that the pulsed laser in the in focus mode caused the heightened DRM.

## Laser-induced plasma of CH$_4$ and CO$_2$

The free electrons can be accelerated by bremsstrahlung absorbing energy of pulsed laser, which increases the electron density like a cascade. Until they have enough energy to collide and ionize the surrounding gas to generate gas plasma, which can be carried out even at

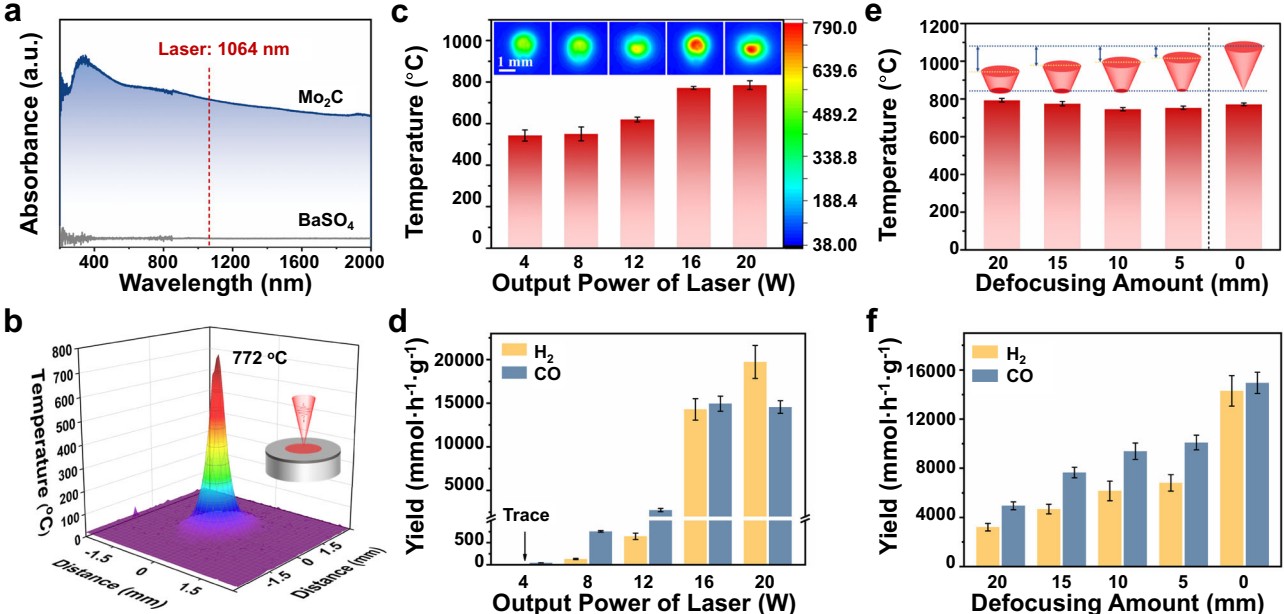

**Fig. 3 | Localized high temperature and corresponding DRM performance under different laser irradiation modes. a** UV-Vis-NIR absorption spectra of $Mo_2C$ and $BaSO_4$. **b** Temperature distribution of laser irradiation on $Mo_2C/BaSO_4$ tablet. **c** The temperatures and **d** the laser-catalytic DRM performances of the $Mo_2C/BaSO_4$ tablet by different pulsed laser powers (4 W, 8 W, 12 W, 16 W, 20 W). Error bars represent standard deviation. **e** The temperatures and **f** the laser-catalytic DRM performances of the $Mo_2C/BaSO_4$ tablet by pulse laser with different defocusing amounts. Error bars represent standard deviation.

low pulsed laser power[48,49]. It is plausible that this laser-induced plasma effect could potentially influence the DRM reaction. Figure 4a presents the excitation spectra of plasmas generated during the DRM process under three conditions: Infocus-Pulsed Laser, Infocus-CW Laser, and Underfocus-Pulsed Laser. With an in focus pulsed laser applied to the $Mo_2C/BaSO_4$ tablet, distinct peaks in the range of 350 to 604 nm corresponding to $CO_2$ and $CH_4$ plasma were detected[30]. Conversely, no plasma generation on $BaSO_4$ under pulsed laser in focus mode confirmed the weak interaction between laser and $BaSO_4$ (Supplementary Fig. 21). In contrast, neither $CO_2$ nor $CH_4$ plasma was observed under CW laser infocus mode (Fig. 4c) or pulsed laser under focus mode (Fig. 4d). Only the envelope peak of thermal radiation (600 - 800 nm) was detected.

The time-dependent dynamic spectra generated under pulsed laser in infocus mode were showed in Fig. 4b. The plasma of $CH_4$ and $CO_2$ were swiftly generated within 50 ms, implying the splitting bremsstrahlung process by pulsed laser. In addition, the plasma intensities were increased with the pulsed laser irradiation time, and peak positions remained relatively stable, confirming the steady generation of plasma by the pulsed laser. Notably, the intensity of plasma increased as the output power of pulsed laser escalated from 4 W to 20 W in infocus mode (Supplementary Fig. 22). In contrast, no spectra of $CH_4$ and $CO_2$ plasma were detected under the CW laser in infocus mode (Fig. 4c) and the pulsed laser in underfocus mode (Fig. 4d). Only the continuous spectrum emitted by atoms at thermodynamically high temperature was observable, which was stable with time. Under the action of CW laser, excited electrons continuously collide with the lattice and emit phonons, converting laser energy into thermal effects and further diffusing into the crystal through heat transfer. While the input energy of the pulsed laser is discontinuous, the collision of excited electrons with the lattice is not sufficient and more electrons can escape from the lattice to become the initial electrons of the cascade reaction, which is necessary to induce avalanche ionization to produce plasma[50-52]. Therefore, the infocus pulsed laser is a pivotal requirement for plasma generation.

A two-dimensional axial symmetry physical model of plasma at the gas-solid interface produced by pulsed laser was established in Fig. 4e. The simulated spatial density distributions of electrons, $^\cdot CH$, and $^\cdot CO$ produced by pulsed laser were shown in Fig. 4f, g, and h. It could be inferred that the generation process of laser induced plasma was as following: Firstly, the hot carriers were generated through interaction between laser and $Mo_2C$. Secondly, hot carriers on $Mo_2C$ absorbed the focused laser energy by a non-resonant process of inverse harsh radiation, which caused avalanche ionization[53], and further cracked $CH_4$ and $CO_2$ molecules to the $^\cdot CH$ plasma and $^\cdot CO$ plasma, respectively.

According to the excitation spectra of laser-induced plasma of $CH_4$ and $CO_2$, the corresponding plasma images were also captured by a high-speed CMOS (Complementary Metal Oxide Semiconductor) camera equipped with band-pass filters centered at 430 nm and 450 nm (Fig. 4i), which recorded the time-resolved shadowgraphs of plasma expansion ($^\cdot CH$ at 431.4 nm, $^\cdot CO$ at 451.1 nm), respectively. As shown in Fig. 4j, for the $Mo_2C$, the plasma plume was captured under pulsed laser in the infocus mode, but no capture occurred under the underfocus mode of pulsed laser or the infocus mode of the CW laser.

As shown in Fig. 4k, the $H_2/CO$ yields of laser-catalytic DRM in CW laser infocus mode and pulsed laser underfocus mode were 5099.8/8000.6 mmol $h^{-1}$ $g^{-1}$ and 6160.8/9389.7 mmol $h^{-1}$ $g^{-1}$, respectively, at similar thermal effect temperatures (Fig. 3e and Supplementary Fig. 23). In contrast, the synergy of laser-induced plasma and laser-induced thermal effects achieved in pulsed laser infocus mode not only doubled the yields but also promoted $H_2$ production.

The above results confirmed that the plasmonization effect induced by the focused pulsed laser was the dominant factor in accelerating the DRM catalytic activity. The synergy between laser-induced plasma and laser-induced thermal effects on $Mo_2C$ contributes to its exceptionally high catalytic activity for DRM. Pulsed laser produces a high-temperature thermal region on the surface of $Mo_2C$, which is thermodynamically sufficient to drive the DRM reaction. Furthermore, the interaction between pulsed laser and $Mo_2C$ generates high-energy electrons, which in turn induce the plasmaization of $CH_4$ and $CO_2$ at the

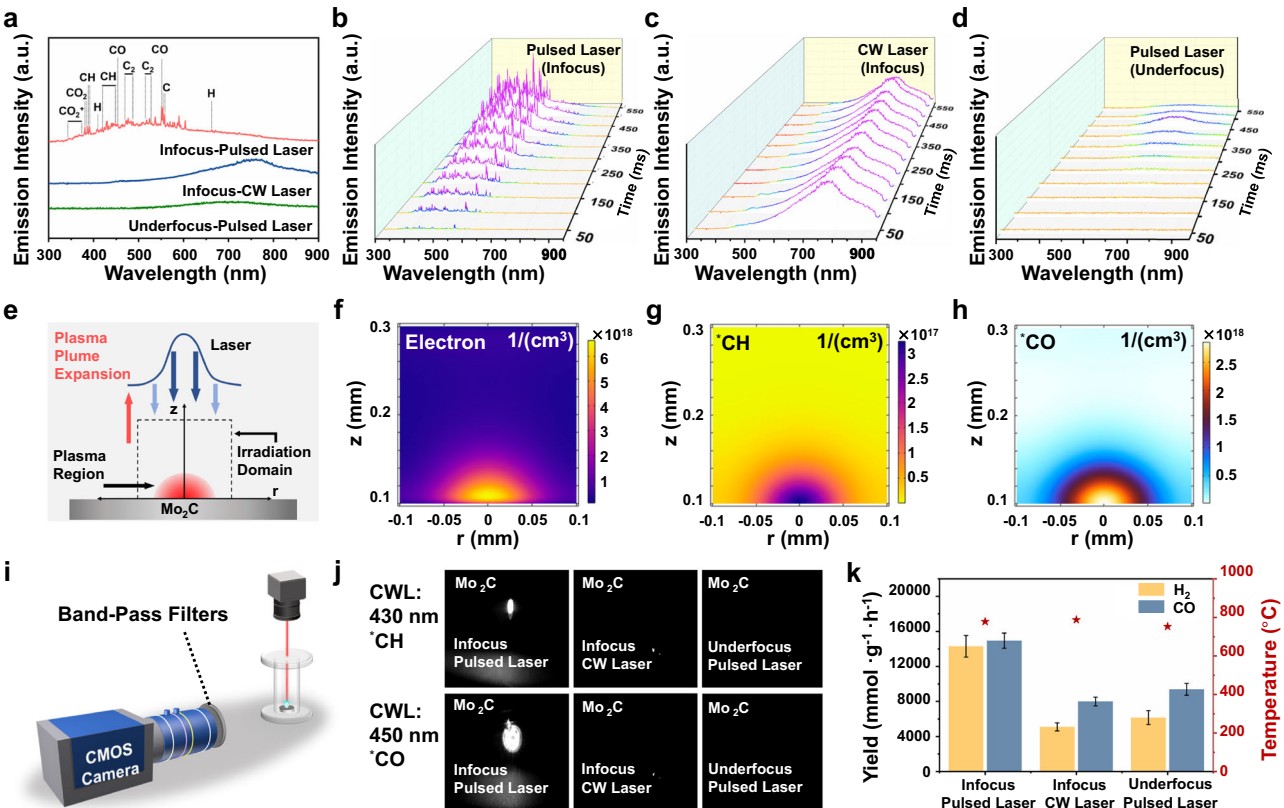

**Fig. 4 | Demonstration of laser-induced plasma. a** Excitation spectra of laser-catalytic DRM under different catalytic conditions. Time-dependent dynamic spectra generated by **b** pulsed laser (Infocus), **c** CW laser (Infocus), and **d** pulsed laser (Underfocus). **e** Schematic diagram of the pulsed laser induced plasma on Mo$_2$C. simulated spatial density distributions of **f** electrically-charged particles including electron, **g** $^*$CH, and **h** $^*$CO of pulsed laser induced plasma. **i** Schematic diagram of high-speed camera equipped with band-pass filter to take plasma optical pictures. **j** The plasma images captured by the high-speed camera equipped with band-pass filters (central wavelength: 430 nm and 450 nm). **k** Performance of laser-catalytic DRM under different catalytic conditions. Error bars represent standard deviation.

gas-solid interface to enhance the DRM activity. In contrast, the CW laser solely triggers the thermal effect on Mo$_2$C due to electron–phonon relaxation, resulting in poorer DRM performance and stability (Supplementary Fig. 24) under the same temperature condition.

Mo$_2$C was an excellent catalyst for CO$_2$ activation during DRM, but its capacity for CH$_4$ activation is relatively weak[21,35]. The schematic mechanism of DRM is shown in Supplementary Fig. 25. The production of C$^*$ requires a four-step process of stepwise dehydrogenation of CH$_4$, which is the rate-limiting step on Mo$_2$C. In conventional thermocatalytic DRM, the reaction potential for generating C$^*$ is higher than that for O$^*$ formation, creating a C$^*$-deficient environment that hampers the structural stability of Mo$_2$C. This scenario renders Mo$_2$C prone to oxidation, ultimately transforming into MoO$_2$ (as illustrated in Supplementary Fig. 26b), which is consistent with previous work[20,54]. The pulsed laser-induced plasma breaks the limiting step of dehydrogenation of CH$_4$, resulting in enhanced catalytic activity (Fig. 2d) and improved stability (Supplementary Fig. 26a) for DRM using a Mo$_2$C catalyst.

### Laser-catalytic DRM in flow system

A flow-type catalytic system is the predominant method for evaluating DRM performance[55]. Consequently, to validate the practical significance of laser catalysis, we established a flow-type laser-catalytic DRM system. (Fig. 5a, b and Supplementary Fig. 27a). Despite laser catalysis being localized to a specific point, the rapid movement of the pulsed laser, with a speed of 1000 mm s$^{-1}$, surpasses the cross-section velocity (0.67 mm s$^{-1}$) of the CH$_4$/CO$_2$ gases (Fig. 5c). This design allows the moving pulsed laser to act as a steady laser line, traversing the reactor chamber vertically along the gas flow direction. The laser line's

rapid oscillation within the reactor maintained the methane conversion of DRM reaction at a stable value of 50.5 %. It is worth noting that the pure Mo$_2$C with poor intrinsic catalytic activity as a catalyst showed relatively poor DRM catalytic performance in the thermocatalytic system[18,32]. Compared to the thermocatalytic system (15.7 %, H$_2$/CO ≈ 0.46, Supplementary Fig. 27b), not only a higher conversion rate (50.5 %) but also a higher H$_2$/CO ratio (H$_2$/CO ≈ 0.86) was implemented in laser-catalytic DRM (Fig. 5d and e). This further affirms the laser's potential to enhance methane cracking, thereby significantly improving DRM's activity and stability. The structural stability of Mo$_2$C after laser-catalysis in flow system was confirmed by XRD pattern (Supplementary Fig. 28), XPS (Supplementary Fig. 29) and HRTEM results (Supplementary Fig. 30), which also demonstrated the above conclusion. Hence, the application of laser-catalytic DRM holds promising potential.

Energy efficiency was also one of the pivotal indicators of different DRM catalytic systems, which were compared in Fig. 5f. Obviously, laser-catalytic DRM demonstrated superior energy efficiency (0.98 mmol kJ$^{-1}$) and more cost-effective electricity conversion (46.8 mmol kW$^{-1}$ h$^{-1}$) compared with recently reported experimental conditions of thermocatalysis[18] and photocatalysis[10], signifying its practical application potential. Moreover, from an industrial perspective, industrial-grade medium-power nanosecond lasers are priced at less than $10,000. It is highly automated and easy to regulate with a stable laser output performance and a long working life, generally up to 100,000 hours. Meanwhile, the reactor used for laser-catalysis is simple in structure and does not need to be subjected to high temperature and pressure during the reaction process. The surface

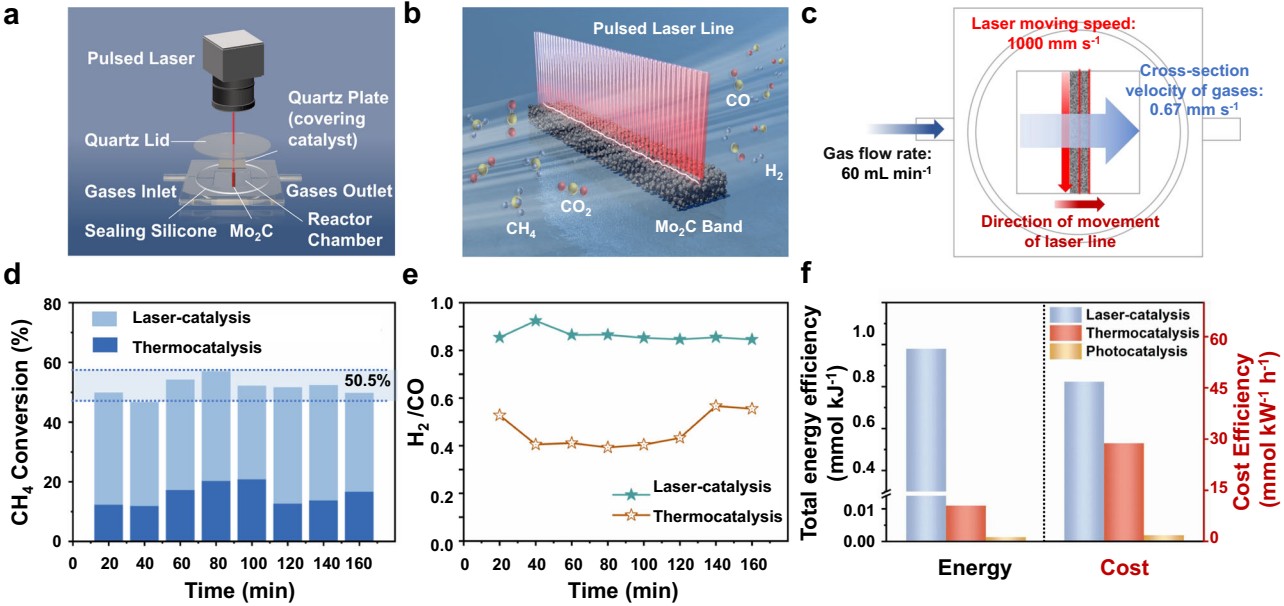

**Fig. 5 | Performance and energy consumption evaluation of laser-catalytic DRM in flow system. a** The laser-catalytic DRM device in flow system. **b, c** Schematic diagram of the interaction among gas molecules, $Mo_2C$ and pulsed laser. Comparison of the **d** $CH_4$ conversion and **e** selectivity of laser-catalytic DRM (16 W) and thermocatalytic DRM (800 W, 800 °C) via the same quantity of $Mo_2C$, Gas velocity: 60 mL $min^{-1}$. **f** Total energy efficiency and the cost efficiency of different catalytic systems.

temperature of the reactor during the laser-catalytic DRM reaction was close to room temperature and very mild due to the thermal effect of the laser was localized (Supplementary Fig. 3b and Supplementary Fig. 32). Furthermore, the catalyst for laser-catalytic DRM is the single $Mo_2C$, easily and inexpensively scalable for large-scale production. Hence, the future application prospects for laser catalysis as a novel catalytic system are highly promising.

## Discussion

A pulsed laser-catalytic DRM reaction was demonstrated without external heating, using simple $Mo_2C$ both as the laser carrier and catalyst. The 16 W pulsed laser induced both thermal and plasma effects on $Mo_2C$ simultaneously in the infocus mode. This innovative approach enhanced the activation capacity of $CH_4$ through laser-induced plasmaization, effectively breaking the rate-limiting step of DRM. Consequently, the laser-catalytic DRM exhibited an exceptionally high catalytic activity, yielding $H_2$ (14300.8 mmol $h^{-1}$ $g^{-1}$) and CO (14949.9 mmol $h^{-1}$ $g^{-1}$), respectively. Simultaneously, an equilibrium reaction between $CO_2$ and $CH_4$ was facilitated by the enhanced laser-induced cracking of $CH_4$, creating a balanced environment that prevented $Mo_2C$ from over-oxidation. This laser-catalytic approach enabled efficient DRM without relying on an external heating source, a breakthrough with significant implications for advancing the research landscape of photothermal catalytic systems.

## Methods
### Synthesis of $Mo_2C$
Firstly, $MoO_2$ nanosheets (NSs) as precursors were prepared by a chemical vapor reduction process in a long quartz tube. Briefly, the phase transition reaction from commercial $MoO_3$ to $MoO_2$ NSs occurred in the existence of Ar−$H_2$ (10% $H_2$) mixture (200 mL $min^{-1}$) at 900 °C for 2 h. After the reaction cooled down naturally, $MoO_2$ NSs were collected at the tail end of the chemical vapor deposition (CVD) system. Due to the equilibrium between the sublimation of $MoO_3$ and the reduction of gaseous $MoO_3$ by Ar-$H_2$, the collected $MoO_2$ NSs exhibit uniformly dispersed hexagonal nanosheet structures of a few microns in size with smooth surfaces and edges. Secondly, the $Mo_2C$

NSs were then prepared by carbonization of $MoO_2$ NSs in a tube furnace. Typically, the obtained $MoO_2$ NSs were placed in a ceramic crucible in the tube furnace and heated up to 1000 °C under Ar (50 mL $min^{-1}$) atmosphere. Once the 1000 °C reached, the Ar gas was shut off and $CH_4$ gas (50 mL $min^{-1}$) was introduced into the tube for 30 mins. Then, the tube furnace was cooled to room temperature naturally under Ar atmosphere. The powders were collected and subsequently washed with ethanol to obtain dark gray $Mo_2C$ NSs.

### Characterization
Phase compositions of the as-made materials were measured by D8 Advance (Germany Bruker) X-ray diffractometer (XRD) with Cu Kα radiation (λ = 0.15406 nm). Morphologies and element mapping of the materials were identified by a field emission scanning electron microscope (SEM, Zeiss, Gemini 300 ; EDS, Oxford, X-Max$^N$ 50) and a transmission electron microscope (TEM, a JEM-2100F Field Emission Electron Microscope, JPN) at an acceleration voltage of 200 kV. X-ray photoelectron spectroscopic (XPS) measurement was performed using a PHI X-tool instrument (Ulvac-Phi). UV-Vis-NIR absorption spectra of samples were recorded by a UV-Vis-NIR spectrophotometer (UH4150, Hitachi High-Technologies Corporation). The production rates of hydrogen and carbon monoxide were measured by a gas chromatograph (GC, Agilent 7890B). High-speed video camera (FuHuang Agile-Device Co., Ltd, X213) was used to verify the plasma plume induced by laser. Time-resolved transmittance spectra were conducted on a home-made device with a portable spectrometer (Aurora 4000, GE-UV-NIR, Changchun New Industries Optoelectronics Tech. Co., Ltd) with step size of 1 nm and dwell time of 50 ms. Infrared thermal images were captured by an infrared thermal imager (Magnity, MAG32HT).

### Pulsed laser-catalytic dry reforming of methane in closed system
Before the laser-catalytic DRM, the synthesized $Mo_2C$ NSs (20 mg) and $BaSO_4$ (0.5 g) were pressed into a catalyst tablet (diameter 13 mm) by a tablet press at a pressure of 100 bar, the catalyst tablet was composed of a thin layer of $Mo_2C$ on the surface of $BaSO_4$ substrate. The $Mo_2C$/$BaSO_4$ tablets with different areas were prepared by laser cutting,

including 0.79, 3.14, 7.07, and 12.57 mm², as shown in Supplementary Fig. 4. Then, by placing the $Mo_2C/BaSO_4$ tablet on the bottom of the quartz reactor (Supplementary Fig. 3a), the mixed gas of $CO_2$, $CH_4$ and Ar at a ratio of 47.5%:47.5%:5% was poured into the reactor for 15 mins to remove air in the reactor, then closed the reactor vent. During the laser-catalytic process, a fiber optic laser system (LSF20D, Hgtech laser) with a laser wavelength of 1064 nm and a pulse duration of 100 ns was used to irradiate the catalyst tablet. A computer was connected to the laser system and the software (named EzCad2) was used to set the experimental parameters and map the catalytic reaction zone. A linear scanning mode with a repetition rate of 20 kHz, maximum power of 20 W, scanning spacing of 0.01 mm and scanning speed of 500 mm s$^{-1}$ was used to perform the laser-catalysis. In order to investigate the effects of laser power and underfocus/infocus on the DRM performance, different powers (4, 8, 12, 16, and 20 W) and different distance of underfocus (defocusing amount = 0, 5, 10, 15, and 20 mm) were utilized during laser-catalysis. Without special explanation, the laser catalytic DRM performances were all obtained in 16 W laser power and infocus laser condition by pulsed laser. The post-reaction gases were analyzed using a GC to obtain the relative amounts of CO, $H_2$, $CO_2$, and $CH_4$.

CW laser-catalytic DRM was used as a control system to confirm the contribution of the pulsed laser-induced plasma effect. CW laser-catalysis induced a high temperature (789 °C) similar to that of the pulse laser-catalysis through 1064 nm CW laser. The catalysts ($Mo_2C/BaSO_4$ tablet), quartz reactor and reaction gas ($CO_2$:$CH_4$:Ar = 47.5%:47.5%:5%) used were consistent with the pulse laser-catalysis system.

### Pulsed laser-catalytic dry reforming of methane in flow system

The laser-catalytic DRM of $Mo_2C$ NSs in a flow system was carried out in a quartz reactor at room temperature and atmospheric pressure. The DRM activity was evaluated under reactive gas flow ($CO_2$:$CH_4$:Ar = 16.7%:16.7%:66.6%). Gas hourly sp/ace velocity (GHSV) with 120 mg $Mo_2C$ catalyst was controlled at 30 L·g$_{cat}$$^{-1}$·h$^{-1}$. A linear scanning mode with a repetition rate of 20 kHz, single pulse energy of 0.8 mJ, scanning spacing of 0.05 mm and scanning speed of 1000 mm s$^{-1}$ was used to perform the laser-catalytic DRM.

The thermocatalytic DRM of $Mo_2C$ NSs in a flow system was carried out in a fixed bed quartz reactor at atmospheric pressure. The activity was evaluated at 800 °C with the same catalyst amount and total flow rate of the feed gas ($CO_2$:$CH_4$:Ar = 16.7%:16.7%:66.6%, GHSV = 30 L g$_{cat}$$^{-1}$ h$^{-1}$) as those of laser-catalysis.

## Data availability

The authors declare that the data supporting the findings of this study are available within this article and its Supplementary Information file, or from the corresponding authors upon request. Source data are provided in this paper.

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

## Acknowledgements

This work was supported by Taishan Scholar Project of Shandong Province (W.J.Z.), Natural Science Foundation of Shandong Province (ZR2021JQ15 to W.J.Z.), ZR2022YQ42 to L.L.Z., ZR2020QE057 to Z.L., and 2022GJJLJRC-01 to W.J.Z.), Innovative Team Project of Jinan (2021GXRC019 to W.J.Z.), the National Natural Science Foundation of China (51972147 to W.J.Z., 52022037 to W.J.Z., and 22172183 to X.W.L.).

## Author contributions

W.J.Z., D.M., and L.L.Z. designed the study. Y.L. synthesized the catalysts. Y.L. and L.L.Z. performed reaction experiments. T.W. and X.Z.Z. performed optical characterization. H.C.H., X.Y.L., Y.K.C., Z.F.T., Z.L., Y.H.Z., and H.L. helped with the discussion. Y.L. and X.W.L. analyzed all the experimental data. Y.L., X.W.L., L.L.Z., D.M., and W.J.Z., wrote the manuscript. All authors interpreted the data and contributed to the preparation of the manuscript.

## Competing interests

The authors declare no competing interests.
