## [Peer Review File · Nature Communications]

Pulsed Laser Induced Plasma and Thermal Effects on Molybdenum Carbide for Dry Reforming of MethaneREVIEWER COMMENTS

Reviewer #1 (Remarks to the Author):

The manuscript focuses on pulsed laser-induced catalytic dry reforming of methane. This technology activates both CH₄ and CO₂ molecules, leading to accelerated DRM reaction activity. The direct generation of CH plasma from CH₄ is attributed to a unique mechanism involving thermionic emission and cascade ionization on the Mo₂C surface. This article provides a valuable overview of pulse laser-based dry methane reforming. However, for publication in this esteemed journal, the authors should address the following comments:

1. The authors need to explain how the thickness of Mo₂C affects the efficiency.
2. The authors must include more experimental evidence (XRD is not sufficient) to support the claim that a C-O balanced environment is created over the catalyst surface during pulse laser catalytic DRM. Can C in Mo₂C be replaced with C of CH₄? Is a carbon deposition/coking observed?
3. What is the scientific reason to select BaSO₄? For examples, inert oxides such as Al₂O₃, MgO or BaO would be the first choice.
4. In the pulse-laser system, is the metallic carbide more plausible rather than metallic catalysts such as Ni? If so, any reasons?
5. The irradiated area is limited in the pulse-laser system. Therefore, the XRD/XPS after tests would be less effective. TEM observation using FIB pick-up system from the irradiated areas would be better to know the changed chemical compositions by the reaction.

Reviewer #2 (Remarks to the Author):

Li et al. have investigated pulsed laser-induced plasma for catalytic dry reforming of methane using Mo₂C. While the authors attempt to explain the pulsed laser effect on dry reforming, the experimental setup and analysis data do not provide sufficient scientific conviction. It cannot be concluded that the plasma effectively addresses the DRM issue such as coking, and the novelty of the material is lacking (<https://www.nature.com/articles/s41467-020-18721-0>) This approach seems to be similar to merely raising the temperature using light (<https://pubs.rsc.org/en/content/articlelanding/2020/cc/d0cc00729c>). Ultimately, due to an apparent oversell of the catalyst's performance compared to reality, this may lead to a misunderstanding among the community. Given the numerous questions in the main text, it is not suitable for publication in Nature Communications. Detailed reasons are provided below.

1. The authors calculated the catalyst amount based on the density (approximately 0.15 mg/mm²) when producing pellets using Mo₂C, referencing the traces investigated by the laser (Supplementary Table 1). However, according to Fig. 3b and c, the thermal region reaching nearly 770 °C extends beyond 2 mm. This suggests that areas damaged by the laser were not the only ones participating in the catalytic reaction; unexamined parts also contributed. Hence, it appears that there is no significant difference in the synthesis gas production rate up to 3.14-12.57 mm², as seen in Fig. 2a. It seems that an insufficient amount of catalyst was considered, leading to data over-exaggeration, as evident in Fig. 2g. Therefore, I believe that all data compared in units of mmol/h/g are erroneous.
2. The scale bars in Fig. 1f and 1i do not seem to match the descriptions. When looking at the EDS image, the thickness of Mo₂C appears to be 40 micrometers. It should be verified whether the calibration is incorrect or if the original (raw) data should be attached for reevaluation. Also, the SEM image in Fig. 1i appears to be more like 0.75 mm than 1 mm. Would it be possible to provide a more accurate scale using an optical microscope?
3. Supplementary Figure 4 suggests that the significant production of CO at temperatures exceeding 800 °C is due to reactions such as Mo₂C + 5CO₂ ⇌ 2MoO₂ + 6CO (<https://www.sciencedirect.com/science/article/pii/S0272884219325349>). This suggests that laser-catalytic DRM may form Mo-O structures as the surface oxidizes (Supplementary Figure 11). Therefore, it does not seem feasible to discuss the stability based on a reaction of less than 8 hours. The catalyst stability should be assessed through at least 400-500 hours of reaction.
4. The most confusing aspect while reading the paper is the closed reactor system (Fig. 1g) and

the flow reactor system (Fig. 5a). Ultimately, the results from the flow reactor system seem more favorable, so why is there a separate explanation in Fig. 2?

5. In Fig. 3a, the authors investigate the absorption wavelength using UV-Vis-NIR before raising the temperature with a 1064 nm laser. Is there a specific reason for choosing the wavelength of 1064 nm? It seems like Mo₂C absorbs more at 400 nm. What results could be predicted if a wavelength of 400 nm were used? Also, lasers are not the cheapest source of energy. How are the authors envisioning industrial applications?

6. In Supplementary Figure 17b, how was the catalyst collected for measurement after the laser catalysis experiment? If BaSO₄ was separated by sulfuric acid treatment, it is necessary to describe how only the post-reaction Mo₂C was collected for XRD measurement, as oxidation would likely occur in the reaction environment (<https://www.nature.com/articles/s41467-020-18721-0>). Continuous DRM will eventually reduce the catalyst's activity. Please clarify this.

7. In Figure captions, provide more detailed descriptions along with subheadings. Additionally, for Figure 4 f-h, make it explicit that they are "simulated" images.

Minor comments:

1. In the main text, line 109, replace "facets" with "planes" as they have different meanings in crystallography.

2. On Supplementary Information page 4, please review the yield calculation. The time unit should be 60 min h⁻¹.

3. Describe experimental conditions in Supplementary Figures 1 and 2.

4. In Fig. 5c, replace the term "velocity of the laser" as it may be confused with the velocity of light (3.0×10^8 m/s). Use a term like "quantity of Mo₂C" instead in Fig. 5e.

Reviewer #3 (Remarks to the Author):

In this manuscript, a DRM reaction on Mo₂C using a pulsed laser was reported. To the best of my knowledge, this approaches for DRM would be new and the performance is good. In addition, the experiments were systematically and well conducted in most parts. However, I have a few concerns and disagreements with some of the interpretations and expressions in the manuscript. If these are properly addressed and corrected, I would be able to recommend acceptance.

[Major points]

1. The data supporting the claim that generated plasma promotes catalytic reactions lack clarity, hindering my understanding. As a premise, I understand that in DRM, if an appropriate metal is used and there is a sufficient reaction temperature, a sufficient reaction rate can be obtained (i.e., the reaction proceeds easily to near chemical equilibrium) in many cases (at least in typical thermal catalysis). I also think that it is generally not easy to measure temperature under light irradiation. Based on these two points, I'd like to confirm the conditions used in this manuscript and judge whether the claim is supported by the experimental data or not.

1a. The authors mentioned "pulse duration of 100 ns". I'd like to confirm the time of the laser off?

1b. If the same laser intensity is used in pulse mode and CW mode, the number of photons input in pulse mode would be lower on average because it contains the off-time. Is my understanding correct? If so, how was the stability in the CW mode in Fig. 2d.

1c. I think "Defocusing" should decrease the maximum temperature because of the decrease in light intensity. However, in Fig. 3e, there is almost no change. What is the reason? Could it be that the maximum local temperature is estimated too low? I am not convinced that the local temperature is 800 °C under the conditions where laser ablation of Mo₂C occurs. It would be better to explain about this point in the manuscript.

1d. Related to 1c, what is the reason why the temperature does not increase when the laser power is increased from 16W to 20W in Fig. 3c?

1e. Am I right in understanding that "Defocusing amount" is the distance shifted from the focal point? In that case, I would like to know the information about the light intensity (I cannot calculate it). I also think that the laser spot size or light intensity is more important to reproducing

the experiment with a different device.

1f. What do the authors think is the reason why the yield increases when the "Defocusing amount" is reduced in Fig. 1f? In addition, is the same phenomenon observed in CW mode? For the reasons I stated in the premise, it is possible that the local heating by the laser will increase the activity compared to larger-area heating. Comparisons of this data with plasma generation amount, etc., may strengthen the claim.

1g. It is reasonable that in this reaction system, an increase in the temperature will increase the reaction rate (and possibly also the selectivity). In this regard, how would the authors remove the possibility that a local increase in temperature might be increasing the reaction rate in the laser system? If this possibility remains, it is difficult to claim the plasma is accelerating the reaction. I believe it is possible that the plasma affects the activity; however, data-based explanations and discussions seem somewhat lacking for understanding.

2. Unfortunately, it is difficult to accept arguments below. Please explain to the comments or revise the manuscript.

2a. In the abstract, the authors wrote, "the limited efficiency of photothermal catalysis", but solar DRM has already been studied with real sunlight in much larger reaction systems (review: 10.1016/j.ijhydene.2015.08.005). What does efficiency mean?

2b. The authors show the catalytic activity per unit weight (g), but the comparison of performance (Supplementary Table 2) should also include the reaction rate (i.e., units in mmol/h, etc.). The high activity in this study is probably due to the very small amount of catalyst used (0.48 mg). (The amount of catalyst used in this study would be less than about 1/10th or less compared to that used in photothermal catalysis studies). The high performance in Fig. 2g may be partially due to the plasma, but it is more likely to be caused by the small amount of catalyst used. In this respect, the comparison in Fig. 2b is not wrong, but it does not mean that this system is superior to the others. At least, the actual production rates should be listed alongside.

2c. In Fig. 2d, what is the reason that CW mode (without plasma) showed a higher performance compared to the thermo catalysis?

2d. In Fig. 2d ("Photothermal catalysis"), it is stated that the low activity is due to weak light and low temperature. If so, this comparison is not fair. 300W Xe lamps would emit light at 15 W or more, and the temperature can reach 800°C or more when one uses a suitable optical setup. Why do the authors use such low-intensity conditions? (It would be better to show the measured temperature in the photothermal conditions).

[Minor point].

3 Significant figures should be taken into account. For example, 14300.8 mmol h⁻¹ g⁻¹.

4 Please check the equations for "total energy efficiency" and "electricity cost". The units seem to be inappropriate.

5 In Fig. 2a and b, the authors describe the horizontal axis as "catalytic area", but I think it is inappropriate. It is not common to think that the irradiated area and the "catalytic area" are the same because the temperature outside the irradiated position also rises.

Responses to the review's comments

We express our sincere gratitude to the reviewers for their valuable and constructive comments on our manuscript. We have addressed each comment in detail and have revised the manuscript thoroughly, considering all the feedback and suggestions. In this response letter, the reviewers' comments are presented in black italics, our responses are in blue, and all changes are marked in red color in the revised manuscript and supporting information.

Reviewer #1 (Remarks to the Author):

The manuscript focuses on pulsed laser-induced catalytic dry reforming of methane. This technology activates both CH₄ and CO₂ molecules, leading to accelerated DRM reaction activity. The direct generation of CH plasma from CH₄ is attributed to a unique mechanism involving thermionic emission and cascade ionization on the Mo₂C surface. This article provides a valuable overview of pulse laser-based dry methane reforming. However, for publication in this esteemed journal, the authors should address the following comments:

1. The authors need to explain how the thickness of Mo₂C affects the efficiency.

Reply: We greatly appreciate the reviewer's good suggestion. Laser-catalytic DRM performances and SEM image of the cross-section of Mo₂C/BaSO₄ tablets with different thickness (such as 36 μm, 54μm, 110μm, 220μm and 315μm) are given in the revised SI (**Supplementary Fig. 6**). Without comparing the mass of the catalysts, the yield of the laser-catalysis system exhibited nearly the same with the gradual increase of Mo₂C thickness. This finding verifies that the thickness of Mo₂C does not affect the yield or selectivity of products in laser catalysis. This is because that the laser is a micro-zone action, whether from the area or depth level, the actual catalytic dose that interacts with the laser to produce thermal effects and plasma effects is very small. In this work, the Mo₂C/BaSO₄ tablet with a thickness of 36 μm was chosen for laser catalysis due to the limitation of the tablet press technology, which is already very thin. We have also added a brief discussion on the thickness of Mo₂C affects the efficiency in the revised text (see page 8, 1st paragraph).

In the revised manuscript, we have added the corresponding description as follows:

“Additionally, we discussed the laser-catalytic DRM performances of Mo₂C/BaSO₄ tablets with different Mo₂C thicknesses (36 μm, 54 μm, 110 μm, 220 μm, and 315 μm). Remarkably, the product rates of H₂ (6.998~6.276 mmol h⁻¹) and CO (7.176~6.201 mmol h⁻¹) did not exhibit significant changes as the Mo₂C thickness increased. This further underscores that the actual catalytic dose interacting with the laser remains very small, regardless of whether it operates at the area or depth level.”

2. The authors must include more experimental evidence (XRD is not sufficient) to support the claim that a C-O balanced environment is created over the catalyst surface during pulse laser catalytic DRM. Can C in Mo₂C can be replaced with C of CH₄? Is a carbon deposition/coking observed?

Reply: We appreciate the valuable comment from the reviewer. In fact, in the original manuscript, we have already provided substantial evidence demonstrating the formation of a C-O balanced environment over the catalyst surface during pulsed laser catalytic DRM. This evidence includes XRD results obtained after the catalytic reaction, along with other supporting data such as the H₂/CO

molar ratio being 1:1 (**Supplementary Fig. 17**), the satisfactory stability observed for the DRM reaction (**Fig. 2e**), and the Mo 3d XPS results for Mo₂C after both laser-catalytic DRM and thermocatalytic DRM (**Supplementary Fig. 19b**). It is widely known that Mo₂C exhibits ideal CO₂ activation capability but is not the dominant factor in methane activation, which mitigates the risk of over-oxidation and eventual deactivation of the catalyst in carbon-deficient environments. Consequently, coke deposition is not readily observed for Mo₂C, as reported in the literature (*ACS Catal.* 2022, 12, 15501–15528). Our comparative experimental results depicted in **Supplementary Fig. 16-18** further validate the effectiveness of laser catalytic DRM as a favorable option. When Mo₂C is utilized for pulsed laser catalytic DRM, the risk of over-oxidation and deactivation is circumvented due to the laser-induced plasma promoting methane cracking. The above statements have been added to the revised manuscript (page 9 paragraph 3 and page 10).

It has been confirmed by multiple studies (*ACS Catal.* 2022, 12, 15501–15528; *Nat. Commun.* 2020, 11,4920) that the carbon in Mo₂C may participate in the DRM reaction. In order to further clarify whether the carbon in Mo₂C is replaced by the carbon in CH₄, we conducted isotopic labeling experiments using ¹³CO₂ and ¹³CH₄. As shown in **Supplementary Fig. 15**, we detected the mass spectra signal of ¹³CO (m/z 29) and ¹²CO (m/z 28). Since Mo₂C maintains a constant stoichiometry during the reaction, this indicates that carbon in Mo₂C exchanges with the reactants, indicating that C in Mo₂C can be replaced with C of CH₄.

To further elucidate the presence of carbon deposition/coking, we have added the Raman spectra of Mo₂C before and after both laser-catalytic and thermocatalytic DRM processes (**Supplementary Fig. 19**). The Raman pattern of the spent Mo₂C catalysts after DRM reveals that fresh Mo₂C exhibits carbon peaks at 1578 cm⁻¹ (G band) and 1345 cm⁻¹ (D band). Subsequent to the thermocatalytic DRM reaction, both the D peak and G peak vanish, indicating a carbon-deficient environment and underscoring the anti-coking properties of Mo₂C. This suggests that there is consumption of carbon and oxidation of Mo₂C occurring during the process.

3. What is the scientific reason to select BaSO₄? For examples, inert oxides such as Al₂O₃, MgO or BaO would be the first choice.

Reply: We thank the reviewer for bringing up this point. In our original text, we have already explained the scientific rationale behind selecting BaSO₄ as the substrate support for the tablet. This choice was made considering that our Mo₂C catalyst requires only about 36 μm in thickness, and BaSO₄ exhibits inertness to both 1064 nm laser radiation and the DRM (**Supplementary Fig.1, Fig.20 and Fig. 21**). The selection of BaSO₄ is based on its physical and chemical properties that make it an effective support material for the catalyst, ensuring that it does not react with any components involved in the catalytic process. Other inert oxides, such as Al₂O₃, MgO, or BaO, share similar properties to BaSO₄ and could serve similar roles (see **Fig. R1**), our selection of BaSO₄ could indeed be substituted with these inert oxides. It also validated the reviewer's first comment that laser is a surface catalysis, which has nothing to do with depth.

Fig. R1 Laser-catalytic DRM performances of Mo₂C tablets with different supports.

4. In the pulse-laser system, is the metallic carbide more plausible rather than metallic catalysts such as Ni? If so, any reasons?

Reply: Many thanks for the reviewer's precious comment which reminded us to deeply consider the applicability of laser-catalysis in different types of catalysts. As the reviewer stated, we think that in the pulse-laser system, the metallic carbide is more plausible rather than metallic catalysts such as Ni. Firstly, First, the ionization potential of CH₄ is lower than that of CO₂, so the activation effect of laser-induced plasma on CH₄ is more significant. It is just that the catalytic property of Mo₂C in DRM reaction possesses that the methane cracking ability is insufficient compared with that of CO₂ activation. Therefore, laser cooperated with Mo₂C in DRM reaction is more conducive to achieving a C-O equilibrium environment. Secondly, the carbon deposition/coking may be slightly reduced due to the activation of CO₂ promoted by laser, but the transition metal itself is easy to catalyze carbon generation, which is also inevitable. Compared with metal catalysts, carbide has the advantage of not easy to carbon deposition. But it presents the limited methane cracking capacity, which causes it to over-oxidize and deactivate easily. Laser-catalysis can make up for the disadvantage of Mo₂C, which promotes methane cracking through thermal emission and avalanche ionization mechanism, producing the C-O equilibrium environment. To sum up, we believe that in the pulse-laser system, the metallic carbide is more plausible rather than metallic catalysts such as Ni.

In order to further verify the above conclusions, we used traditional Ni/MgO catalyst to perform laser-catalytic DRM reaction, and the results were shown in **Fig. R2**. After laser-catalysis for 15mins with Ni/MgO as catalyst, the yield of H₂ product was significantly higher than that of CO, which confirmed that Ni catalyst has excellent methane cracking ability, while CO₂ activation ability is relatively insufficient. This also caused a C-rich environment in the Ni-catalyzed DRM reaction, and carbon deposition occurred. With the increase of reaction time, it can be seen that the product yield decreased significantly, which may be due to the decrease of catalytic activity caused by carbon deposition. The Raman results of the catalyst before and after the DRM reaction also confirmed the existence of carbon deposition in Ni/MgO after laser catalysis. It validates our

conclusion that in the pulse-laser system, the metallic carbide is more plausible rather than metallic catalysts such as Ni.

Fig. R2 (a) Laser-catalytic DRM performances of Ni/MgO. (b) Raman spectra of the fresh Ni/MgO and spent Ni/MgO catalysts. (Pulsed laser-catalytic DRM in closed system: 16W pulsed laser under infocused mode; mass of Ni/MgO=0.48 mg; CO₂:CH₄:Ar = 47.5%:47.5%:5%)

5. The irradiated area is limited in the pulse-laser system. Therefore, the XRD/XPS after tests would be less effective. TEM observation using FIB pick-up system from the irradiated areas would be better to know the changed chemical compositions by the reaction.

Reply: We appreciate the reviewer's concerns regarding the reliability of the XRD and XPS characterization data. In this study, the spent samples used for structure-stability characterization were meticulously separated within the laser irradiation region. For the XRD and Raman characterizations that require a substantial sample volume (arranged in a circular area with a diameter of 5 mm and a height of 0.5 mm), we collected samples approximately 20 times, totaling about 5.3 mg. Thus, our characterization results are reasonable. Following the reviewer's suggestion, we have added TEM images of spent Mo₂C, obtained using a FIB pick-up system from the laser-irradiated areas (**Fig. R3** as **Supplementary Fig. 14**). The HRTEM images post-laser catalysis still exhibited the lattice fringes corresponding to the (101) and (002) planes of Mo₂C. Consequently, the C-O equilibrium environment established by the laser-catalytic DRM reaction effectively prevents the peroxidation of the Mo₂C catalyst. We have also added a brief discussion on the characterization results in the revised text (see page 9, paragraph 2).

Fig. R3 HRTEM image of spent Mo₂C using FIB pick-up system from the laser irradiated areas.

It is worth noting that although the irradiation range of the laser in the intermittent reaction (**Fig. 2**) is limited, but for the laser-catalytic DRM in flow system (**Fig. 5**), the laser is not irradiated in the form of point radiation, line scanning and surface scanning were performed through setting the laser-control program (**Fig. R4**). In order to further verify the stability of the catalyst in laser catalysis, we also supplemented the relevant structural characterization of the catalyst in the laser-catalytic DRM in flow system (with line scanning mode and a large irradiated area). As shown in **Supplementary Fig. 28-30**, the XRD pattern and XPS results revealed that Mo₂C phase remained intact and no serious oxidation occurred. HRTEM image of Mo₂C after laser-catalysis in flow system exhibited the distinct lattice planes of Mo₂C. The results indicated that laser-induced plasma effect in laser catalysis can indeed make up for the insufficient CH₄ cracking ability of Mo₂C, and build the C-O equilibrium to ensure the catalytic and structural stability of Mo₂C, no matter for the laser scanning in small area or for the line scanning in flow system.

Fig. R4. Schematic diagrams of laser irradiation point in closed reaction system and laser line scanning in flow reaction system.

Reviewer #2 (Remarks to the Author):

Li et al. have investigated pulsed laser-induced plasma for catalytic dry reforming of methane using Mo₂C. While the authors attempt to explain the pulsed laser effect on dry reforming, the experimental setup and analysis data do not provide sufficient scientific conviction. It cannot be concluded that the plasma effectively addresses the DRM issue such as coking, and the novelty of the material is lacking (<https://www.nature.com/articles/s41467-020-18721-0>) This approach seems to be similar to merely raising the temperature using light (<https://pubs.rsc.org/en/content/articlelanding/2020/cc/d0cc00729c>). Ultimately, due to an apparent oversell of the catalyst's performance compared to reality, this may lead to a misunderstanding among the community. Given the numerous questions in the main text, it is not suitable for publication in Nature Communications. Detailed reasons are provided below.

Reply: Thank the reviewer for your thoughtful review and critique of our manuscript detailing the use of pulsed laser-induced plasma for catalytic dry reforming of methane using Mo₂C. We appreciate the opportunity to address the concerns raised regarding the experimental setup, data analysis, and the novelty of our approach. We've revised our manuscript and provided additional data to address the concerns raised, offering a clearer perspective on our approach's novelty and effectiveness.

In current work, the key point we proposed is that the two effects of gas plasma and localized high temperature induced by pulsed laser were confirmed by experiments and theories. The introduction of pulsed laser breakthroughs the limited methane cracking capacity and surface overoxidation challenges for Mo₂C in DRM, which was due to that the laser induced plasma activation of the reactants creates a C*-O* balanced environment conducive to maintaining the stability of Mo₂C. The excitation spectra and the plasma images captured by high-speed camera could directly detect laser-induced plasma, and the comparison of catalytic performance between pulsed laser and CW laser confirmed that plasma was crucial to improve the DRM performance besides the thermal effect caused by laser. Both of pulsed laser and CW laser have thermal effects and the same temperature was ensured, but pulsed laser has significantly enhanced DRM activity, confirming the key role of pulsed laser-induced plasma effect.

As the reviewer mentioned "It cannot be concluded that the plasma effectively addresses the DRM issue such as coking", it is well established that Mo₂C possessed ideal CO₂ activation capability, but was not dominant in methane activation, which leads to over-oxidation and eventually the deactivation of the catalyst in carbon-deficient environments. Because of this, coke deposition was not easy to occur for Mo₂C, and this has been reported in the literature (*ACS Catal.* 2022, 12, 15501–15528). However, Mo₂C is easy to be oxidized and has poor catalytic stability. Therefore, various reports on how to improve its CH₄ cracking capacity by modifying Mo₂C have been developed, such as loading Ni (*Appl. Catal. B Environ.* 2022, 301, 120779; *Appl. Catal. B Environ.* 2022, 310, 121250) etc. The loading of metals on Mo₂C can regulate the activation balance of CO₂ and CH₄ to achieve the catalytic stability of DRM (*Chem* 2023, 9, 102-116). It's worth noting that laser-catalytic DRM provides a good choice using simple Mo₂C catalyst without modification. When using Mo₂C for laser catalytic DRM, the over-oxidation and deactivation of Mo₂C were avoided due to the laser-induced plasma promoting methane cracking. The C*-O* balanced environment is established, and this can be proved by XRD results after the catalytic reaction (**Supplementary Fig. 13**), as well as some other evidence, such as the selectivity of H₂/CO

molar ratio with 1:1 (**Supplementary Fig. 17**), the satisfactory stability for DRM reaction (**Fig. 2e** and **Supplementary Fig. 12**), and the Mo 3d XPS results for Mo₂C after laser-catalytic DRM and thermocatalytic DRM (**Supplementary Fig. 19b**). In addition, to further clarify whether there is carbon deposition/coking, we have supplemented the Raman spectra of Mo₂C before and after laser-catalytic and thermocatalytic DRM as well as the TEM images of Mo₂C using FIB pick-up system from the laser irradiated areas. As shown in **Supplementary Fig. 19a**, Raman pattern of the spent Mo₂C catalysts after DRM showed that fresh Mo₂C itself contains carbon peaks at 1578 cm⁻¹ (G band) and 1345 cm⁻¹ (D band). After thermocatalytic DRM reaction, the D band and G band disappear, indicating that a C-deficient environment and the anti-coking advantages of Mo₂C. This resulted in C consumption and oxidation of Mo₂C. While in the laser-catalytic DRM, C-O equilibrium environment leads to neither carbon deposition nor oxidation of Mo₂C. The TEM images of Mo₂C using FIB pick-up system from the laser irradiated areas in **Supplementary Fig. 14** also confirmed that the C-O equilibrium environment constructed by laser-catalytic DRM reaction effectively avoids peroxidation of Mo₂C catalyst. As shown, after laser catalysis, the HRTEM image still maintained the lattice fringes corresponding to (101) and (002) planes of Mo₂C, which was consistent with that before laser catalysis.

As the reviewer mentioned “the novelty of the material is lacking”, we think that the focus of this work is not Mo₂C, but the laser induced thermal effect and plasma effect synergistically enhanced efficiency and stability of Mo₂C catalysts for DRM reaction. Although the Mo₂C catalyst used in this work is relatively common and has been reported widely, it can perfectly confirm the innovation point of this work, that is, the combination of 16W pulsed laser with low-cost unmodified Mo₂C could achieve a significant DRM performance under mild condition, even commercial Mo₂C can be achieved. This is because that the pulsed laser initiates the direct generation of *CH plasma from CH₄ through a thermionic emission mechanism. This mechanism bypasses the conventional step-by-step dehydrogenation of methane, thus eliminating the rate-limiting step in methane cracking. The significant CH₄ activation ability of laser makes up for the advantage of Mo₂C with poor CH₄ cracking ability, which constructing the C*-O* balance environment on the catalyst surface without adding metal sites, which is of great significance to maintaining the stability of the catalyst. This work confirms the existence of laser-induced thermal and plasma effects through experimental data, and demonstrates the reason why pure Mo₂C without any metal-loading or modification has a such high DRM performance under pulsed laser catalysis.

As the reviewer mentioned “This approach seems to be similar to merely raising the temperature using light”, we are sorry that the data analysis descriptions in the manuscript didn’t clearly highlight the critical role of laser-induced plasma effects, besides of thermal effects. The excitation spectra and the plasma images captured by high-speed camera could directly detect laser-induced plasma, and the comparison of catalytic performance between pulsed laser and CW laser confirmed that plasma was crucial to improve the DRM performance besides the thermal effect caused by laser. As shown in **Fig. 4c**, neither CO₂ nor CH₄ plasma was observed under CW laser infocus mode, implying that CW laser solely triggers the thermal effect on Mo₂C due to electron-phonon relaxation. Thus, under the same temperature condition, the yields of the CW laser-driven DRM were substantially lower in comparison to those of the pulsed laser (**Fig. 2d**), confirming the pulsed laser-induced plasma effect plays a key role in laser-catalytic DRM. In addition, for the traditional photothermal catalytic DRM reactions, no *CH or *CO plasmas are detected. Therefore, the performance of laser catalysis has been improved by orders of magnitude compared with that of

photothermal catalysis. Under the conditions of 3 W cm^{-2} irradiation, the performance of photothermal catalytic DRM was re-evaluated using a 300 W Xe lamp with a plano-convex lens. Even when heated to $361 \text{ }^\circ\text{C}$ under 30 solar light intensities (**Supplementary Fig. 8**), the activity of Mo_2C in the photothermal catalytic DRM was only $0.251 \text{ mmol h}^{-1} \text{ g}^{-1}$ (yield of H_2) and $0.824 \text{ mmol h}^{-1} \text{ g}^{-1}$ (yield of CO) (**Fig. 2d**).

As the reviewer mentioned “due to an apparent oversell of the catalyst’s performance compared to reality, this may lead to a misunderstanding among the community.” We are sorry for the insufficient description of experimental details and unclear definitions of terms within our study. A detailed explanation is included in the reply to question 1. In addition, we also compared the DRM performance without comparing the mass of the catalysts, the yield (6.86 mmol h^{-1} for H_2 and 7.18 mmol h^{-1} for CO) of the laser-catalysis system with less catalyst ($0.48 \text{ mg Mo}_2\text{C}$) was still better than that (1.47 mmol h^{-1} for H_2 and 4.87 mmol h^{-1} for CO) of the thermocatalysis system ($50 \text{ mg Mo}_2\text{C}$), suggesting the superior intrinsic catalytic activity of laser-driven DRM.

We hope these revisions and additional data comprehensively address the concerns raised and better articulate the significance and innovation of our work in advancing DRM research.

1. The authors calculated the catalyst amount based on the density (approximately 0.15 mg mm^{-2}) when producing tablets using Mo_2C , referencing the traces investigated by the laser (Supplementary Table 1). However, according to Fig. 3b and c, the thermal region reaching nearly $770 \text{ }^\circ\text{C}$ extends beyond 2 mm . This suggests that areas damaged by the laser were not the only ones participating in the catalytic reaction; unexamined parts also contributed. Hence, it appears that there is no significant difference in the synthesis gas production rate up to $3.14\text{-}12.57 \text{ mm}^2$, as seen in Fig. 2a. It seems that an insufficient amount of catalyst was considered, leading to data over-exaggeration, as evident in Fig. 2g. Therefore, I believe that all data compared in units of $\text{mmol h}^{-1} \text{ g}^{-1}$ are erroneous.

Reply: We greatly appreciate the reviewer’s attention and insightful comments on our work. We have noted that the reviewer’s concerns and misunderstandings regarding the catalytic activity data presented in $\text{mmol h}^{-1} \text{ g}^{-1}$ units stem from insufficient description of experimental details and unclear definitions of terms within our study.

To address this issue, we have clarified certain experimental details for the reviewer’s consideration. In this work, we have defined and distinguished three areas, including Mo_2C tablet area (actual area), thermal area (Fig. 3b-c) and laser irradiation area (**Fig. 1i**).

(a) Tablet area: we have prepared tablets with different areas, including 0.79 mm^2 , 3.14 mm^2 , 7.07 mm^2 and 12.57 mm^2 , as shown in **Fig. R5**. The corresponding total actual Mo_2C mass was 0.12 mg , 0.48 mg , 1.07 mg and 1.91 mg , respectively (**Table R1**). **In our manuscript, we used the total mass of Mo_2C tablet to calculate the mass activity.**

(b) Thermal area: As shown in the inset of **Fig. 3c**, the temperature exceeding $540 \text{ }^\circ\text{C}$ is confined to the central red region with a diameter of approximately 1.1 mm , which constitutes the primary contribution to catalytic performance. The outer green annular region, maintains a temperature of approximately $400 \text{ }^\circ\text{C}$. As shown in **Fig. 3c** and **3d**, when the laser power is 4 or 8 W , the average surface temperature of the catalyst is around $500 \text{ }^\circ\text{C}$, indicating very low catalytic activity. This suggests that the primary contribution to catalysis comes from the high-temperature region with a central diameter of approximately 1.1 mm (area of 0.95 mm^2). We define this area as the thermal

area, and the effective Mo₂C mass in thermal area is calculated to be 0.14 mg. Only for the Mo₂C tablet with the tablet area of 0.79 mm², due to the limitation of the actual tablet area, the thermal area is also 0.79 mm² and the corresponding Mo₂C mass is 0.12 mg (**Table R2**).

(c) **Laser irradiation area:** As shown in the SEM results of Fig. 1i, the diameter of the irradiated region is 0.9 mm with the irradiated area of 0.69 mm², and the corresponding effective Mo₂C mass in irradiated area is calculated to be 0.10 mg (**Table R3**).

For the calculation of catalytic activity, we calculated the mass activity using the Mo₂C mass corresponding to these three areas respectively, and the results are shown in **Fig. R6** and **Table R1-R3**. **The catalytic activity calculated by the total Mo₂C mass in tablet area is experimental activity, while the catalytic activities calculated by the effective Mo₂C masses in thermal area and irradiated area are theoretical activities.** Obviously, the theoretical activity is much higher than the experimental activity for Sample 1-3 with the tablet area of 12.57 mm², 7.07 mm² and 3.14 mm². **In our manuscript, the data we used was the experimental activity of sample 3 with tablet area of 3.14 mm², so we can understand that the activity of laser catalysis is underestimated.**

In addition, we are sorry for the misunderstanding about the inaccurate temperature distribution data in previous Fig. 3b (**Fig. R7a**). The part selected by the solid line box in **Fig. R7b** is the temperature distribution data measured by the infrared camera, and the dotted line part is a schematic diagram of the temperature distribution. In **Fig. R7a**, the correspondence between the measured data and the schematic diagram does not match. The correct correspondence is shown in **Fig. R7b**. In order to avoid misunderstanding, we directly use infrared thermal imaging temperature profile to display the temperature distribution (**Fig. R7c** as **Fig. 3b**). Furthermore, to more effectively present the catalytic data as experimental activity calculated based on the total mass of Mo₂C, we replaced ‘catalytic area’ on the horizontal axis of **Fig. 2a** and **2b** with ‘tablet area’ in the revised manuscript. We have also included a schematic representation of various catalytic testing modes (**Fig. R5** as **Supplementary Fig. 4**) alongside the corresponding catalytic activities (presented in **Fig. R6** as **Supplementary Fig. 5** and **Tables R1-R3** as **Supplementary Tables 1-3**). We trust these modifications and additions will address the reviewer’s concerns and enhance the comprehensibility of our research.

Fig. R5. Schematic representation of various catalytic testing modes. Gray circle represents the catalyst tablet area, Light red circle represents the thermal area of the catalyst tablets, Red circle represents the irradiation area of laser.

Fig. R6. Catalytic activity of the samples using the Mo₂C mass in three areas.

Table R1. Catalytic performance of the samples calculated by total Mo₂C mass within tablet area.

Sample	Tablet Area (mm ²)	Mo ₂ C Mass (mg)	Product rate (mmol/h)		Activity (mmol/h/g)	
			H ₂	CO	H ₂	CO
1	12.57	1.91	7.091	7.611	3712.0	3984.3
2	7.07	1.07	7.330	7.080	6850.5	6616.8

3	3.14	0.48	6.864	7.176	14291.7	14937.5
4	0.79	0.12	4.201	4.802	35000.0	40000.0

Table R2. Catalytic performance of the samples calculated by effective Mo₂C mass within thermal Area.

Sample	Thermal Area (mm ²)	Effective Mo ₂ C Mass (mg) ^a	Product rate (mmol/h)		Activity (mmol/h/g)	
			H ₂	CO	H ₂	CO
1	0.95	0.14	7.091	7.611	50642.9	54357.1
2	0.95	0.14	7.330	7.080	52357.1	50571.4
3	0.95	0.14	6.864	7.176	49000.0	51214.3
4	0.79	0.12	4.201	4.802	35000.0	40000.0

a. Effective Mo₂C Mass is defined as the amount of Mo₂C covered within a thermal area of 0.95 mm².

Table R3. Catalytic performance of the samples calculated by effective Mo₂C mass within irradiation Area.

Sample	Irradiated Area (mm ²)	Effective Mo ₂ C Mass (mg) ^b	Product rate (mmol/h)		Activity (mmol/h/g)	
			H ₂	CO	H ₂	CO
1	0.69	0.10	7.091	7.611	68173.1	73173.1
2	0.69	0.10	7.330	7.080	70480.8	68076.9
3	0.69	0.10	6.864	7.176	65961.5	68942.3
4	0.69	0.10	4.201	4.802	40384.6	46153.9

b. Effective Mo₂C Mass is defined as the amount of Mo₂C covered within a laser-irradiated area of 0.69 mm².

Fig. R7. (a) Previous and (b) modified pictures of temperature distribution of laser irradiation on Mo₂C/BaSO₄ tablet (.). (c) Infrared thermal imaging temperature profile of laser irradiation on Mo₂C/BaSO₄ tablet.

2. The scale bars in Fig. 1f and 1i do not seem to match the descriptions. When looking at the EDS image, the thickness of Mo₂C appears to be 40 micrometers. It should be verified whether the

calibration is incorrect or if the original (raw) data should be attached for reevaluation. Also, the SEM image in Fig. 1i appears to be more like 0.75 mm than 1 mm. Would it be possible to provide a more accurate scale using an optical microscope?

Reply: We are grateful for the reviewer's insightful observation regarding the scale bars in Fig. 1f and 1i. In response to the concerns raised, we have taken the opportunity to re-examine and accurately re-dimension these figures based on the original data scales (We have attached the original data for **Fig. 1e**, **Fig. 1f**, and **Fig. 1i** as **Fig. R8-10**). We have also updated the manuscript to reflect these revised measurements accurately. This action ensures the integrity of our data presentation and supports the clarity and accuracy of our findings.

Fig. R8 Original SEM image of the cross-section of Mo₂C/BaSO₄ tablet.

Fig. R9 Original EDS mapping image of the cross-section of Mo₂C/BaSO₄ tablet.

Fig. R10 Original SEM image of the circular spot of Mo₂C/BaSO₄ tablet irradiated by pulsed laser.

3. *Supplementary Figure 4 suggests that the significant production of CO at temperatures exceeding 800 °C is due to reactions such as $\text{Mo}_2\text{C} + 5\text{CO}_2 \rightarrow 2\text{MoO}_2 + 6\text{CO}$ (<https://www.sciencedirect.com/science/article/pii/S0272884219325349>). This suggests that laser-catalytic DRM may form Mo-O structures as the surface oxidizes (Supplementary Figure 11). Therefore, it does not seem feasible to discuss the stability based on a reaction of less than 8 hours. The catalyst stability should be assessed through at least 400-500 hours of reaction.*

Reply: We deeply appreciate the valuable suggestions and insightful inquiries, especially regarding the long-term evaluation of catalyst stability.

The stability of Mo₂C in DRM reactions is a concern due to its tendency to oxidize, leading to deactivation. It's been reported that Mo₂C has excellent CO₂ activation capability but lacks in CH₄ activation, which can result in over-oxidation (*ACS Catal.* 2022, 12, 15501–15528). To counteract this, modifications such as loading Ni on Mo₂C have been explored to improve methane cracking capacity and achieve a balance in CO₂ and CH₄ activation, thus enhancing catalytic stability (*Appl. Catal. B Environ.* 2022, 301, 120779; *Appl. Catal. B Environ.* 2022, 310, 121250). In our experiment, we also confirmed that the Mo₂C is unstable and easily oxidized, resulting in unstable catalytic activity in thermocatalytic DRM process (**Supplementary Fig. 19** and **26**).

Laser-catalytic DRM offers an innovative approach by using unmodified Mo₂C and leveraging the laser-induced plasma to promote methane cracking, thereby avoiding over-oxidation and deactivation of Mo₂C. This method establishes a C*-O* balanced environment, which is supported by various pieces of evidence, including XRD results after the catalytic reaction (**Supplementary Fig. 13**), as well as some other evidence, such as the selectivity of H₂/CO molar ratio with 1:1 (**Supplementary Fig. 17**), the satisfactory stability for DRM reaction (**Fig. 2e** and **Supplementary Fig. 12**), and the Mo 3d XPS results for Mo₂C after laser-catalytic DRM and thermocatalytic DRM (**Supplementary Fig. 19b**).

Therefore, if C-O equilibrium cannot be achieved in the DRM reaction, only one of the cases "such as coking" or "form Mo-O structures as the surface oxidizes" proposed by the reviewer can occur. For Mo₂C, the problem of "form Mo-O structures as the surface oxidizes" in the traditional thermocatalysis will occur, mainly because that Mo₂C has a good activation capacity of CO₂, while

CH₄ activation capacity is insufficient. In the laser-catalysis process, the laser-induced plasma effect solves the insufficient CH₄ activation of Mo₂C, which realizing the C-O equilibrium reaction and then achieving a high catalytic stability. For the Mo XPS spectra in **Supplementary Fig. 19b**, laser-catalytic DRM showed a slight increase in the peaks of higher-valence Mo states compared to unreacted molybdenum carbide, but the thermocatalytic DRM possessed the obvious increase in peaks of higher-valence Mo states.

We acknowledge the reviewer's emphasis on the necessity of long-term stability assessments for a thorough evaluation of catalyst performance. While we concur with this perspective, we face practical constraints in our laboratory setup that impede our ability to conduct evaluations over the recommended span of 400-500 hours. Despite these limitations, we have successfully completed a 50-hour stability test, as depicted in **Fig. R11** (as **Supplementary Fig. 12**), where the catalyst demonstrated commendable performance and stability. These initial findings provide a promising indication of the robustness of our catalytic system, even though they fall short of the reviewer's suggested duration. We are grateful for the reviewer's constructive critique and keen interest in our study.

Fig. R11 Stability testing of laser-catalytic DRM using a Mo₂C/BaSO₄ tablet for 50 h.

4. The most confusing aspect while reading the paper is the closed reactor system (Fig. 1g) and the flow reactor system (Fig. 5a). Ultimately, the results from the flow reactor system seem more favorable, so why is there a separate explanation in Fig. 2?

Reply: The DRM reaction in flow reactor system seem more favorable. However, in this work, the main innovation is that we experimentally verified that the mechanism of laser combining with Mo₂C to improve the DRM reaction efficiency, which is synergistic effect of laser-induced thermal effect and plasma effect. The temperature characterization of the thermal effect and the emission spectrum characterization of the plasma effect are consistent with the laser irradiation conditions in the closed reactor system. The flow reactor system cannot realize the characterizations of the laser-induced temperature and plasma, etc., so it is not conducive to the study the interaction between laser and Mo₂C. Therefore, there is a separate explanation in **Fig. 2** and **Fig. 5**. We first studied the mechanism of laser-catalytic DRM reaction through closed reactor system, and then verified the practicability through flow reactor system.

5. In Fig. 3a, the authors investigate the absorption wavelength using UV-Vis-NIR before raising the temperature with a 1064 nm laser. Is there a specific reason for choosing the wavelength of

1064 nm? It seems like Mo₂C absorbs more at 400 nm. What results could be predicted if a wavelength of 400 nm were used? Also, lasers are not the cheapest source of energy. How are the authors envisioning industrial applications?

Reply: Many thanks for the reviewer's comment and suggestion. In this work, we chose 1064 nm laser for two reasons: Firstly, DRM reaction is a thermodynamic endothermic reaction, which needs to reach a certain temperature to drive the reaction. As a near-infrared wavelength laser, 1064 nm laser possesses more obvious thermal effect than ultraviolet laser. we have added the temperature characterization and performances of Mo₂C in 355 nm laser catalysis. As shown in **Fig. R12**, the maximum temperature induced by the 355 nm laser on the surface of Mo₂C is only about 330 °C, which cannot drive the DRM reaction, and we did not detect the H₂ and CO products by GC. Although the UV-Vis-NIR absorption spectrum shows that Mo₂C has a stronger absorption at 300-400 nm, due to the higher photon energy of 355 nm laser. Secondly, compared to ultraviolet laser (Laser wavelength: 355 nm, Pulse duration: 100 ns, Price: \$35,000), 1064 nm laser (Laser wavelength: 1064 nm, Pulse duration: 100 ns, Price: \$9,693) is low-cost and technological maturity.

As the reviewer's concern, lasers are not the cheapest source of energy, but the overall use cost of laser is markedly decreasing, creating good prospects for its wide application in various fields. The cost of using the laser includes the purchase cost and power consumption.

(a) Purchase cost: The current price of KW-level lasers on the market is only thousands of dollars, which has a price advantage compared with traditional processing tools and equipment. In this work, the utilized industrial-grade medium-power nanosecond lasers are priced at less than \$10,000. It is highly automated and easy to regulate with a stable laser output performance and a long working life, generally up to 100,000 hours.

(b) Power consumption: With the advancement of laser technology, the current most advanced lasers can achieve more than 50% electro-optical conversion efficiency (such as: the United States IPG company's YLS-ECO laser, electro-optical conversion efficiency of 50%), effectively reducing production costs. In this work, we compared the energy efficiency of different DRM catalytic systems, as shown in **Fig. 5f**. Obviously, laser-catalytic DRM demonstrated superior energy efficiency (0.98 mmol kJ⁻¹) and more cost-effective electricity conversion (46.8 mmol kW⁻¹ h⁻¹) compared with recently reported experimental conditions of thermocatalysis and photocatalysis, signifying its practical application potential.

Meanwhile, the reactor used for laser-catalysis is simple in structure and does not need to be subjected to high temperature and pressure during the reaction process. The surface temperature of the reactor during the laser-catalytic DRM reaction was close to room temperature and very mild due to the thermal effect of the laser was localized (**Supplementary Fig. 3b**). Furthermore, the catalyst for laser-catalytic DRM is the single Mo₂C, easily and inexpensively scalable for large-scale production. Hence, the future application prospects for laser catalysis as a novel catalytic system are highly promising.

Fig. R12 Temperature distribution on the surface of the Mo₂C/BaSO₄ tablet under focused irradiation with 355 nm pulse laser.

6. In Supplementary Figure 17b, how was the catalyst collected for measurement after the laser catalysis experiment? If BaSO₄ was separated by sulfuric acid treatment, it is necessary to describe how only the post-reaction Mo₂C was collected for XRD measurement, as oxidation would likely occur in the reaction environment (<https://www.nature.com/articles/s41467-020-18721-0>). Continuous DRM will eventually reduce the catalyst's activity. Please clarify this.

Reply: Many thanks for the reviewer's comment and suggestion.

In original **Supplementary Fig. 17b**, the pulsed laser-catalytic DRM in flow system with 120 mg Mo₂C powders were performed. In the flow system, there is no tablet pressing and no use of BaSO₄, so the Mo₂C after laser-catalytic reaction is directly collected. As for the closed reaction system, after the laser catalysis experiment, the spent samples used for structure-stability characterization were carefully separated in the laser irradiation region and without sulfuric acid treatment. For the XRD and Raman characterization that requires a large amount of samples (laid out in a $\phi 5 \times 0.5$ mm circular area), we have collected about 20 times and 5.3 mg. So the characterization results are real and reliable. The XRD patterns (**Supplementary Fig. 13**) revealed that Mo₂C phase remained intact under CO₂: CH₄ feed ratio of 1:1 atmosphere. In addition to XRD results, other characterizations that do not require extensive powders, such as XPS (**Supplementary Fig. 19b**) and TEM (**Supplementary Fig. 14**), can also prove the structural stability of the Mo₂C after laser catalysis.

As mentioned in the reviewer and cited literature, Mo₂C will decrease its activity due to oxidation during the long-term DRM reaction, which is because Mo₂C itself cannot balance the activation of CH₄ and CO₂, and this is also consistent with the thermocatalytic results in this work (**Supplementary Fig. 16-18**). The promotion effect of laser-induced plasma on CH₄ cracking is significant, which effectively inhibits the formation of O*-rich environment on the surface of Mo₂C, and a higher stability of the catalyst are obtained. This is the core innovation of laser catalysis.

Regarding the evaluation of catalyst stability over long periods, we understand and agree with the reviewer's point that assessing the stability through extended reaction times is crucial for a comprehensive understanding of catalyst performance. However, due to limitations in laboratory conditions and experimental duration constraints, conducting stability evaluations over 400-500 hours presents a significant challenge for us at this moment. Nevertheless, we have completed a 50-

hour stability test (**Supplementary Fig. 12**), during which the catalyst exhibited good performance and stability. These preliminary results offer positive evidence for our catalytic system, although the test duration did not meet the length suggested by the reviewer. We thank the reviewer again for their valuable feedback and attention to our work. We look forward to further guidance and suggestions from the reviewer to help us improve and refine our research.

7. *In Figure captions, provide more detailed descriptions along with subheadings. Additionally, for Figure 4 f-h, make it explicit that they are “simulated” images.*

Reply: We would thank for the reviewer’s valuable comments and suggestions. According to the reviewer’s suggestion, all the captions of **Fig.1-Fig.5** have been supplemented more detailed descriptions along with subheadings in the revised manuscript, and the captions of **Fig. 4 f-h** have been revised as “simulated spatial density distributions of (f) electrically-charged particles including electron, (g) *CH , and (h) *CO of pulsed laser induced plasma.” In addition, the laser-induced plasma was verified in **Fig. 4a, 4b, 4j** by experimental means.

Minor comments:

1. *In the main text, line 109, replace "facets" with "planes" as they have different meanings in crystallography.*

Reply: Thank you for the reviewer’s suggestion. We have checked thoroughly and replaced "facets" with "planes" in the revised manuscript.

2. *On Supplementary Information page 4, please review the yield calculation. The time unit should be 60 min h^{-1} .*

Reply: We appreciate your meticulous attention in identifying the discrepancy in the time unit used for the yield calculation. We sincerely apologize for this oversight and any confusion it may have caused. The necessary correction has now been implemented on page 4 of the Supplementary Information, ensuring the time unit is accurately represented as 60 min h^{-1} . This amendment aligns with the correct unit conversion, thereby clarifying any ambiguity and maintaining the integrity of our data presentation.

The reaction yields were calculated based on the mass of the Mo_2C :

$$\text{Yield (mmol}\cdot\text{h}^{-1}\cdot\text{g}^{-1}) = \frac{|\Delta p(\%)| \cdot V \text{ (mL)} \cdot 60 \text{ (min}\cdot\text{h}^{-1})}{t \text{ (min)} \cdot m_{Mo_2C} \text{ (g)} \cdot 22.4 \text{ (mmol}\cdot\text{mL}^{-1})}$$

3. *Describe experimental conditions in Supplementary Figures 1 and 2.*

Reply: We are grateful for the reviewer’s suggestion. We have added detailed descriptions of the experimental conditions in Supplementary Figures 1 and 2, improving manuscript clarity and quality.

4. *In Fig. 5c, replace the term "velocity of the laser" as it may be confused with the velocity of light ($3.0 \cdot 10^8 \text{ m/s}$). Use a term like "quantity of Mo_2C " instead in Fig. 5e.*

Reply: We appreciate the reviewer's attention to detail and their suggestion for clarity in our terminology. As recommended, we have amended **Fig. 5c** to read "laser spot moving speed" instead of "velocity of the laser" to prevent any potential confusion with the speed of light. Additionally, in **Fig. 5e**, we have updated the term to "quantity of Mo₂C" to more accurately reflect the content of the figure.

Reviewer #3 (Remarks to the Author):

In this manuscript, a DRM reaction on Mo₂C using a pulsed laser was reported. To the best of my knowledge, this approaches for DRM would be new and the performance is good. In addition, the experiments were systematically and well conducted in most parts. However, I have a few concerns and disagreements with some of the interpretations and expressions in the manuscript. If these are properly addressed and corrected, I would be able to recommend acceptance.

[Major points]

1. The data supporting the claim that generated plasma promotes catalytic reactions lack clarity, hindering my understanding. As a premise, I understand that in DRM, if an appropriate metal is used and there is a sufficient reaction temperature, a sufficient reaction rate can be obtained (i.e., the reaction proceeds easily to near chemical equilibrium) in many cases (at least in typical thermal catalysis). I also think that it is generally not easy to measure temperature under light irradiation. Based on these two points, I'd like to confirm the conditions used in this manuscript and judge whether the claim is supported by the experimental data or not.

Reply: We would thank for the reviewer's precious comments and suggestions.

As stated by the reviewer, if an appropriate metal is used and there is a sufficient reaction temperature, a sufficient reaction rate can be obtained (i.e., the reaction proceeds easily to near chemical equilibrium) in many cases (at least in typical thermal catalysis). It is indeed, but the catalyst used in this work is Mo₂C, the nature of Mo₂C is that the methane cracking capacity in thermal catalysis is limited, and it is easy to be oxidized and deactivated in DRM reaction. In laser catalysis, the plasma effect generated by laser promotes the methane cracking and produces a C-O equilibrium environment, so that the DRM reaction rate is significantly improved and the reaction reaches chemical equilibrium. In theory, DRM can occur when there is a sufficient reaction temperature, such as with a metal catalyst. However, during the DRM process, the carbon deposition reaction will occur on the surface of the metal catalyst, and the catalytic stability and activity will be deteriorated. Therefore, a lot of work has been done to modify the catalyst so that the catalyst can have both activation capacity of CO₂ and CH₄ to achieve C-O equilibrium reaction. In this work, different from the traditional catalyst modification, we use the simple Mo₂C as a catalyst, the laser-induced plasma effect solves the insufficient CH₄ activation of Mo₂C, which realizing the C-O equilibrium reaction and then achieving a high catalytic stability. It is a new idea to improve DRM activity, this manuscript uses Mo₂C as verification, and in the future work, we will also verify the interaction mechanism of other catalysts and laser to enhance DRM activity.

As the reviewer's concern, it is generally not easy to measure temperature under light irradiation. Although the temperature test of the catalyst surface in laser catalysis may be not accurate, in this work, all the laser test conditions use the same infrared thermal imaging test method, even if the specific value of the temperature tested may not be accurate, its trend is certain. That is, the temperature is similar (**Fig. R13** as Supplementary Fig. 23), but the significant performance difference in performance between the infocusing pulsed laser, infocusing CW laser in **Fig. 4k** indicates that the plasma effect generated by laser is the main mechanism to enhance the DRM reaction of Mo₂C.

In order to more clearly highlight the role of laser-induced plasma, we reprocessed the tested temperature induced by CW laser and presented it in the same way as that of pulsed laser. As shown in **Fig. R13**, it is noteworthy that pulsed laser and CW laser have the same thermal effect with

consistent temperatures. Therefore, the difference in performance is attributed to the plasma effect induced by the pulsed laser. In addition, we also added the test conditions of CW laser-catalytic DRM in the experimental section, and added the corresponding temperatures of infocusing pulsed laser, infocusing CW laser and the underfocusing pulsed laser in Fig. 4k. Try to describe the data and present the graph in a way that makes it clearer to the reader.

In the supplementary information, we add the following description:

“Continuous wave (CW) laser-catalytic DRM was used as a control system to confirm the contribution of the pulsed laser-induced plasma effect. CW laser-catalysis induced a high temperature (789 °C) similar to that of the pulse laser-catalysis through 1064 nm CW laser. The catalysts (Mo₂C/BaSO₄ tablet), quartz reactor and reaction gas (CO₂:CH₄:Ar = 47.5%:47.5%:5%) used were consistent with the pulse laser-catalysis system.”

Fig. R13 The temperature generated by (a) 16W pulse laser focused irradiation and (b) CW laser irradiation on the Mo₂C/BaSO₄ tablet.

1a. The authors mentioned "pulse duration of 100 ns". I'd like to confirm the time of the laser off?

Reply: We are thankful for the reviewer’s inquiry regarding the pulse duration and the inter-pulse interval. To clarify, the pulse duration of 100 ns refers to the active emission time of the laser pulse. This duration encompasses the rise and fall times of the photons from half-maximum to peak intensity and back to half-maximum, as illustrated in Fig. R14.

The pulse period, on the other hand, is the total time between the start of one pulse and the start of the next pulse. In our experiments, we utilized a laser with a frequency of 20 kHz, which translates to a pulse period of 50 μs (0.00005 seconds). Therefore, the time interval or the “laser off” time between two consecutive pulses is the pulse period minus the pulse duration, which would be slightly less than 50 μs, considering the pulse duration is 100 ns.

In mathematical terms, the frequency (f) is the reciprocal of the period (T), thus: $f = 1/T$

For a frequency (f) of 20 kHz: $T = \frac{1}{f} = \frac{1}{20000} = 50 \times 10^{-6} \text{ s} = 50 \mu\text{s}$

Subtracting the pulse duration (100 ns) from the pulse period (50 μs) gives us the “laser off” time:

$$\text{"Laser off" time} = T - \text{pulse duration} = 50 \mu\text{s} - 100 \text{ ns}$$

This calculation provides the time interval between the end of one pulse and the beginning of the next pulse. We hope this addresses your question and aids in understanding the laser operation parameters used in our study.

Fig. R14 Schematic diagram of (a) Laser pulse train at 20 kHz and (b) the pulse duration in a single pulse.

1b. If the same laser intensity is used in pulse mode and CW mode, the number of photons input in pulse mode would be lower on average because it contains the off-time. Is my understanding correct? If so, how was the stability in the CW mode in Fig. 2d.

Reply: We would thank for the reviewer's precious comments and suggestions. As stated by the reviewer, if the same laser intensity is used in pulse mode and CW mode, the number of photons input in pulse mode would be lower on average because it contains the off-time. This understanding is true under certain conditions, and only applies when pulsed lasers are implemented through an external modulator, As shown in **Fig. R15**. There are many ways to realize the pulse laser, the nanosecond laser pulse in this work is realized by the internal modulator (Q switch). The basic principle is to allow the laser pumping process to produce a particle population inversion far beyond the typical threshold, which can prevent laser oscillations, while ensuring a large cavity loss. Optical feedback suppression is achieved by adding loss in the laser cavity. After a large inversion is achieved, the intracavity feedback is turned on again. The laser then experiences a gain that greatly exceeds the loss, and the stored energy is released in ultra-short, high-intensity pulses of light. As a result, the number of photons input in pulse mode was higher on average than that in CW mode. Each pulse has a great peak power, and the pulse energy is very high. This is also the key to pulsed laser can produce plasma effect and CW laser cannot achieve.

In this work, in order to verify the plasma effect, we regulate the power of the CW laser to ensure that the CW laser has the same temperature to the pulse laser. Because the CW laser only produces thermal effect, without plasma effect, the CO proportion in the CW laser product is higher than H₂, and the overall product yield is low. Under the synergistic action of thermal effect and plasma effect, pulsed laser produces excellent DRM activity and stability.

According to the reviewer's suggestion, we have supplemented the stability measurement in the CW mode. As shown in **Fig. R16** (as **Supplementary Fig. 24**), the CO proportion in the CW laser product is higher than H₂, and the overall product yield is low. This is because that the CW laser only produces thermal effect, without plasma effect. Due to the limited CH₄ cracking capacity, Mo₂C is easily oxidized and deactivated during CW laser-catalysis, so the stability results showed that the performance began to decline after 30 mins of CW laser-catalysis, and the Mo₂C catalyst was almost completely deactivated after 60 mins. This result confirms our conclusion above, only

pulsed laser simultaneously produces thermal effect and plasma effect to promote methane cracking, the stable catalysis of Mo_2C is achieved.

Fig. R15 Schematic diagram of the output energy comparison between pulsed laser output achieved using an external modulator (a) and an internal modulator (b).

Fig. R16. The catalytic stability of Mo_2C in CW laser-catalytic DRM.

1c. I think "Defocusing" should decrease the maximum temperature because of the decrease in light intensity. However, in Fig. 3e, there is almost no change. What is the reason? Could it be that the maximum local temperature is estimated too low? I am not convinced that the local temperature is $800\text{ }^\circ\text{C}$ under the conditions where laser ablation of Mo_2C occurs. It would be better to explain about this point in the manuscript.

Reply: We deeply appreciate the reviewer's insightful observations and queries regarding the temperature dynamics observed in our experiments. Firstly, it needs to be clear that the laser output of the infocusing laser and the defocusing laser is the same, but the infocusing laser focuses within a very small region, and the defocusing laser irradiation region increases. Secondly, the average temperature within a certain region was obtained by the infrared thermal imager. In Fig. 3e, we think that the approximate temperature of infocusing laser and defocusing laser on Mo_2C surface is caused by the following reasons: The transformation of photon energy involves two aspects when laser interacting with Mo_2C . On the one hand, phonon energy exchange occurs between photon and

the Mo₂C lattice to induce the thermal effect. On the other hand, the interaction between laser and Mo₂C excites initial hot electrons for subsequent avalanche ionization to induce the plasma effect. Under defocusing laser, the irradiation region is increased, resulting in a small photon energy per unit area, which can only make a larger range of temperature region heating, the plasma effect is weak. While for the infocusing laser, due to the focus on a very small region, the photon energy per unit area is very high, this causes a small area to heat up and is accompanied by plasma activation. Therefore, from the point of view of energy conservation, it is reasonable that the infocusing laser and the defocusing laser produce a similar temperature, because part of the infocusing laser energy is used to produce plasma.

In terms of the laser local temperature measurement, the infrared thermal images are captured by an infrared thermal imager (Magnity, MAG32HT) in this work. The highest measured temperature can reach 3000 °C, which eliminates the problem of inaccurate temperature measurement caused by the infrared thermal imager reaching the upper temperature limit. In order to further verify the measured temperature value, we also use 0.6mm diameter tungsten-rhenium thermocouple to measure (**Supplementary Table 6**), the measured temperature is basically consistent with that detected by the infrared thermal imager. Of course, the local temperature of the laser is currently difficult to measure accurately, which is a problem for the industry. The current temperature test is the measurement of the average temperature within a certain region, the actual local temperature value may be different from the measured temperature, but we are unified test conditions and test equipment, to ensure that the temperature trend is accurate.

In response to your feedback, we have included additional discussion in the manuscript to clarify these points and provide a more comprehensive understanding of the temperature dynamics under different laser focusing conditions. We hope this addresses your concerns and enhances the clarity of our findings.

In the revised manuscript, we have supplemented the corresponding discussion about the laser local temperature as follows (see page 11, 1st paragraph):

“Of course, the local temperature of the laser is currently difficult to measure accurately, which is a problem for the industry. The current temperature test is the measurement of the average temperature within a certain region, the actual local temperature value may be different from the measured temperature, but we are unified test conditions and test equipment, to ensure that the temperature trend is accurate.”

Supplementary Table 6 Laser-induced temperature under different laser irradiation conditions.

Laser irradiation conditions	Temperature (°C)	
	infrared thermal imager	tungsten-rhenium thermocouple
20 W pulsed laser; infocus	784	760
16 W pulsed laser; infocus	771	759
12 W pulsed laser; infocus	619	599
8 W pulsed laser; infocus	550	532
4 W pulsed laser; infocus	542	519
16 W; underfocus (DA = 5 mm)	753	742
16 W; underfocus (DA = 10 mm)	746	751
16 W; underfocus (DA = 15 mm)	775	789
16 W; underfocus (DA = 20 mm)	793	787
CW laser	789	791

1d. Related to 1c, what is the reason why the temperature does not increase when the laser power is increased from 16W to 20W in Fig. 3c?

Reply: Thanks for the reviewer's precious comment. As we replied in question 1c, the transformation of photon energy involves two aspects when laser interacting with Mo₂C. On the one hand, phonon energy exchange occurs between photon and the Mo₂C lattice to induce the thermal effect. On the other hand, the interaction between laser and Mo₂C excites initial hot electrons for subsequent avalanche ionization to induce the plasma effect. Under the same pulse duration, a higher average power (for example, 20W) of pulsed laser means that it has a higher peak power. When the peak power of laser reaches a certain value, the photon energy is high enough to excite the initial hot electrons when interacting with Mo₂C for subsequent avalanche ionization to produce plasma effect. At low peak power, phonon energy exchange predominates to produce thermal effect. Therefore, we believe that 16W average power is the threshold power for laser and Mo₂C interaction to produce a large amount of plasma, and the higher photon energy at 20W has little effect on the thermal effect, while the plasma effect is stronger, the excitation spectra of pulsed laser-catalytic DRM at different output powers in **Supplementary Fig. 22** demonstrated this conclusion.

1e. Am I right in understanding that "Defocusing amount" is the distance shifted from the focal point? In that case, I would like to know the information about the light intensity (I cannot calculate it). I also think that the laser spot size or light intensity is more important to reproducing the experiment with a different device.

Reply: Thanks for the reviewer's precious comments. Yes, "Defocusing amount" is the distance shifted from the focal point. The infocusing laser focuses within a very small region, as shown in **Fig. R17** (as **Supplementary Fig. 31**) with a spot diameter of 100 μm. When we gradually change the defocusing amount, the spot diameter gradually increases, and the corresponding single pulse energy density gradually decreases (**Supplementary Table 5**). As mentioned, when laser interacting with Mo₂C, the transformation of photon energy involves two aspects including the thermal effect and the plasma effect. With the decrease of single pulse energy density, the plasma effect cannot be realized and only exhibits the thermal effect. Thus, the performances of laser-catalytic DRM reaction is significantly affected. According to the reviewer's suggestion, we have supplemented the schematic diagram of laser beam focusing and the spot area and single pulse energy density of laser at different defocusing amount in the revised supporting information.

Fig. R17 Schematic diagram of laser beam focusing.

$$E = \frac{P_{\text{avg}}}{f_{\text{rep}}} = \frac{16 \text{ W}}{20 \text{ kHz}} = 0.8 \text{ mJ}$$

$$P_{\text{peak}} = \frac{P_{\text{avg}}}{f_{\text{rep}} \cdot \tau} = \frac{16 \text{ W}}{20 \text{ kHz} \cdot 100 \text{ ns}} = 8 \text{ kW}$$

$$\text{Single pulse energy density} = \frac{E}{S} \text{ (mj cm}^{-2}\text{)}$$

where E is the energy of a single pulse, P_{avg} is the average output power of the laser, f_{rep} is the repetition frequency, τ is the pulse width, and P_{peak} is the peak power, S is spot area.

Supplementary Table 5 Spot area and single pulse energy density of laser at different defocusing amount.

Defocusing Amount (mm)	Spot Area (mm ²)	Single pulse energy density (mJ/cm ²)
0	0.008	10000
5	0.080	1000
10	0.221	362
15	0.442	181
20	0.739	108
25	1.111	72

If. What do the authors think is the reason why the yield increases when the "Defocusing amount" is reduced in Fig. 1f? In addition, is the same phenomenon observed in CW mode? For the reasons I stated in the premise, it is possible that the local heating by the laser will increase the activity compared to larger-area heating. Comparisons of this data with plasma generation amount, etc., may strengthen the claim.

Reply: We would thank for the reviewer's precious comments and suggestions. In terms of the reason why the yield increases when the "Defocusing amount" is reduced in Fig. 1f, it is obviously that with the decrease of the defocusing amount, the spot diameter gradually increases, corresponding gradually increased optical power density. So we think that the increase in yield is related to the increased optical power density. Just as replied in question 2c, CW laser catalysis and

thermocatalysis are both cases where no plasma plays a role, and only thermal effects affect the DRM reactivity. The difference is that CW laser generates local heat, while traditional thermocatalysis achieves overall heating through electric heating. We think that local heat may indeed be conducive to photothermal catalysis. This conjecture plays a very enlightening role for our future research.

We believe that in the laser-catalysis, the laser-induced plasma effect is the key to achieving an efficient DRM activity, and the temperature is only the basic condition to ensure that DRM reaction occurs. The main purpose of controlling the defocusing amount is to achieve the plasma generation. As shown in **Fig.4 a, j** and **k**, the results demonstrate that plasma can be produced only under laser infocusing mode, and only when plasma is generated can high catalytic activity be obtained. If only high temperatures are achieved without plasma production, high catalytic activity cannot be obtained. Therefore, the plasma is the main reason for the high catalytic activity, not the temperature.

According to the reviewer's suggestion, we have supplemented the temperature distribution and catalytic DRM performances of the Mo₂C/BaSO₄ tablet under focused CW laser with different defocusing amounts. As shown in **Fig. R18** and **R19**, different from pulsed laser mode, the surface temperature of Mo₂C decreases with the increasing of defocusing amount in CW laser mode. This is consistent with our conventional acknowledge (the more focused the light, the higher the heat temperature), and precisely shows that CW laser has no plasma effect, only thermal effect. The pulsed laser action has both thermal effect and plasma effect. Under infocusing condition, because part of the laser energy is used to produce plasma effect, the infocusing temperature can be approximately maintained with the defocusing temperature. The DRM properties were also consistent with the CW laser induced temperature and thermocatalytic properties, which also confirmed the above conclusions.

According to the above results, the infocused pulsed laser and infocused CW laser have the same thermal effect with the same thermal area (0.95 mm²) and consistent temperatures (772 °C for infocused pulsed laser and 789 °C for infocused CW laser) (**Supplementary Fig. 23**). However, the significant performance difference (**Fig. 4k**) between the infocused pulsed laser (with plasma) and infocused CW laser (without plasma) indicates that the plasma effect generated by laser is the main mechanism to enhance the DRM reaction of Mo₂C instead of thermal effect (**Fig. 4a** and **4j**).

Fig. R18 Temperature distribution on the surface of the $\text{Mo}_2\text{C}/\text{BaSO}_4$ tablet under focused CW laser with different defocusing amounts.

Fig. R19 The laser-catalytic DRM performances of the $\text{Mo}_2\text{C}/\text{BaSO}_4$ tablet by CW laser with different defocusing amounts.

Ig. It is reasonable that in this reaction system, an increase in the temperature will increase the reaction rate (and possibly also the selectivity). In this regard, how would the authors remove the possibility that a local increase in temperature might be increasing the reaction rate in the laser system? If this possibility remains, it is difficult to claim the plasma is accelerating the reaction. I believe it is possible that the plasma affects the activity; however, data-based explanations and discussions seem somewhat lacking for understanding.

Reply: Thanks for the reviewer's precious comments.

The comparison between the pulsed laser-catalysis and the CW laser-catalysis at similar temperatures is the most direct experimental data (**Supplementary Fig. 23**), which can demonstrate that the plasma effect induced by pulsed laser significantly enhances the DRM performance (**Fig. 4k**). It is noteworthy that pulsed laser and CW laser have the same thermal effect with consistent temperatures. The catalytic DRM activity of pulsed laser is significantly higher than that of CW laser, which confirms that the pulsed laser induced plasma effect plays a key role in improving DRM activity by excluding the local temperature increase.

In addition, the effect of different defocusing amounts on laser-catalytic DRM performance also confirmed that the thermal effect was not the sole driver of the enhanced performance of laser-catalytic DRM. The temperatures remained relatively consistent in both underfocus (defocusing amount = 20, 15, 10, 5 mm) and infocus modes (defocusing amount = 0 mm), ranging from 746 ~ 793 °C (**Fig. 3e** and **Supplementary Table 6**), the laser-catalytic DRM performances exhibited significant discrepancies, as shown in **Fig. 3f**. In the infocus mode, the H₂/CO yields of 14300.8/14949.9 mmol h⁻¹ g⁻¹ were significantly higher than those in the underfocus modes (3212.8/4953.934, 4681.0/7654.8, 6813.9/10094.0 and 6160.8/9389.7 mmol h⁻¹ g⁻¹). These results confirmed that the pulsed laser in the infocus mode caused the heightened DRM catalytic activity. Furthermore, the H₂/CO molar ratio was significantly increased with the generation of plasma, also confirms that pulsed laser boosted the plasma activation of CH₄.

For clarity, we have added the test conditions of CW laser-catalytic DRM in the experimental section, and added the corresponding temperatures of infocusing pulsed laser, infocusing CW laser and the underfocusing pulsed laser in **Fig. 4k**, which trying to describe the data and present the graph in a way that makes it clearer to the reader.

In the supplementary information, we also add the following description:

“Continuous wave (CW) laser-catalytic DRM was used as a control system to confirm the contribution of the pulsed laser-induced plasma effect. CW laser-catalysis induced a high temperature (789 °C) similar to that of the pulse laser-catalysis through 1064 nm CW laser. The catalysts (Mo₂C/BaSO₄ tablet), quartz reactor and reaction gas (CO₂:CH₄:Ar = 47.5%:47.5%:5%) used were consistent with the pulse laser-catalysis system.”

2. Unfortunately, it is difficult to accept arguments below. Please explain to the comments or revise the manuscript.

2a. In the abstract, the authors wrote, "the limited efficiency of photothermal catalysis", but solar DRM has already been studied with real sunlight in much larger reaction systems (review: 10.1016/j.ijhydene.2015.08.005). What does efficiency mean?

Reply: We would thank for the reviewer's precious comments and suggestions. We are sorry for the improper expression. We have revised the expression in the abstract according to the reviewer's suggestion. As the reviewer mentioned, solar DRM has already been studied with real sunlight in much larger reaction systems. At present, photothermal catalysis has made important progress in DRM reactions, especially focusing solar DRM, which mainly focuses on photothermal conversion and catalytic site regulation. In this paper, a high photothermal conversion temperature is generated by using laser as the light source, and the reaction gas can be plasmonized to realize efficient dry reforming. Importantly, the requirement for catalytic active sites in laser-catalysis is not high, and even ordinary Mo₂C can be used as a catalyst. For the problem of C*-O* imbalance caused by

insufficient Mo₂C catalytic site, laser catalysis solves it through laser-induced plasma effect. Of course, Laser catalysis may achieve better performance if used with catalysts that have been reported to have better performance in photothermal catalysis or in thermal catalysis.

In the abstract, we have revised as follows:

“Dry reforming of methane (DRM) is highly endothermic reaction. Harsh reaction conditions of thermocatalysis hinders its development. Here we introduce a novel approach for DRM reaction utilizing a 16 W pulsed laser as a low-energy source in conjunction with a cheap Mo₂C catalyst, enabling the realization of DRM under milder reaction conditions.”

The literature the reviewer suggested is very helpful in broadening our understanding of photothermal catalysis. We have cited this literature as ref [6] in the revised manuscript.

[6] Elysia, J.S., Esmail, M.A.M., Ahmed, F.G., A review of solar methane reforming systems. *Int. J. Hydrogen. Energ.* **40**, 12929-12955 (2015).

2b. The authors show the catalytic activity per unit weight (g), but the comparison of performance (Supplementary Table 2) should also include the reaction rate (i.e., units in mmol/h, etc.). The high activity in this study is probably due to the very small amount of catalyst used (0.48 mg). (The amount of catalyst used in this study would be less than about 1/10th or less compared to that used in photothermal catalysis studies). The high performance in Fig .2g may be partially due to the plasma, but it is more likely to be caused by the small amount of catalyst used. In this respect, the comparison in Fig. 2b is not wrong, but it does not mean that this system is superior to the others. At least, the actual production rates should be listed alongside.

Reply: We deeply appreciate the reviewer’s valuable suggestions and concerns regarding the impact of catalyst loading on the estimation of catalytic activity. In response to your advice, we have added data on reaction rates in the Supplementary Information, Table S4, to provide a more comprehensive performance comparison. In addition, in order to confirm that the high performance in **Fig. 2g** is not caused by the small amount of catalyst used, but the dominant role of laser-induced plasma, we separately supplemented the following comparative experiments:

Firstly, the same small amount of Mo₂C catalysts (**0.48 mg**) were used for thermocatalysis and laser catalysis, respectively. As shown in the performance comparison (**Fig. R20** as **Supplementary Fig. 11**), under the same small amount, the product rate of H₂ (0.009 mmol h⁻¹) and CO (0.037 mmol h⁻¹) and yields of H₂ (17.7 mmol h⁻¹ g⁻¹) and CO (77.2 mmol h⁻¹ g⁻¹) in thermocatalysis were obtained, which were much lower than those of laser catalysis (yields of H₂ of 14300.8 mmol h⁻¹ g⁻¹ and CO of 14949.9 mmol h⁻¹ g⁻¹). This confirmed that thermocatalysis could not achieve high catalytic activity under a small amount of catalysts, only laser catalysis can achieve high DRM activity under such a small amount of catalysts (0.48 mg), which is the advantage of laser catalysis.

Secondly, different amounts of catalysts measurements (0.48 mg for laser catalysis and 50 mg for thermocatalysis) were also performed for comparison. As shown in **Fig. 2d** and **Fig. R21** (as **Supplementary Fig. 10**), even without comparing the mass of the catalysts, the yield (6.86 mmol h⁻¹ for H₂ and 7.18 mmol h⁻¹ for CO) of the laser-catalysis system with less catalyst (0.48 mg Mo₂C) was still better than that (1.47 mmol h⁻¹ for H₂ and 4.87 mmol h⁻¹ for CO) of the thermocatalysis system (50 mg Mo₂C), suggesting the superior intrinsic catalytic activity of laser-driven DRM.

More importantly, during the laser catalysis process, laser-induced plasma effect promotes methane activation and cracking, which changes the product selectivity (H_2/CO) from ~ 0.19 in thermocatalysis to ~ 1.1 in laser-catalysis, significantly increasing the H_2 ratio (Fig. R21).

We concur with the reviewer's viewpoint that when comparing different catalytic systems, all relevant parameters should be considered comprehensively. Therefore, we have adjusted our data presentation according to the suggestions to ensure a fair comparison of the performance of each system. It is worth noting that some experimental details in the cited literature are not described in detail. We try our best to compare the reaction rate (units in $mmol/h$) and the catalytic activity per unit weight ($mmol/h/g$). We hope these adjustments and additions adequately address the reviewer's concerns and further demonstrate the efficacy of our system.

Fig. R20 Performances of the laser-catalysis system and thermocatalysis system with the same small amount of catalysts (0.48 mg Mo_2C).

Fig. R21 The production rate of the laser-catalysis system with less catalyst (0.48 mg Mo_2C) and thermocatalysis system (50 mg Mo_2C).

Supplementary Table 4 Performance of some catalysts from recent studies.

Catalyst	Reaction Conditions	Catalyst mass (mg)	Production Rate ($mmol\ h^{-1}$)		Yield ($mmol\ h^{-1}\ g_{cat}^{-1}$)		H_2/CO	Ref
			H_2	CO	H_2	CO		
Mo_2C	Light Source:1064 nm Fiber	0.48	6.864	7.176	14300.8	14949.9	0.98	This

	laser (operating at 16 W); P = 1 atm; No additional thermal input							work
Ni@HZSM-5	Thermocatalysis (CO ₂ :CH ₄ =3.1:1); P = 1 atm; T = 550 °C	200 (Ni 0.85 wt.%)	2.866	5.275	1686 mmol h ⁻¹ g _{Ni} ⁻¹	3103 mmol h ⁻¹ g _{Ni} ⁻¹	-	5
Ni-Mo-MgO	Thermocatalysis; P = 1 atm; T = 800 °C	50	93.750	107.100	1875	2142	0.88	6
Cu-Ru single-atom alloy	Light Source: white light from a supercontinuum laser (19.2 W cm ⁻²); P = 1 atm; No additional thermal input	1.5	2.279	2.295	1519	1530	0.99	7
SCM-Ni/SiO ₂	Light Source: 500 W Xe lamp; P = 1 atm; No additional thermal input	24.3	24.932	29.014	1026	1194	0.86	8
La _{0.9} Ca _{0.1} Fe _x N _{1-x} O _{3-δ}	Thermocatalysis; P = 1 atm; T = 850 °C	200	98.000	102.800	490	514	0.95	9
Ni-Fe Nanoalloy	Light Source: 300 W xenon lamp (3.62 W cm ⁻²); P = 0.18 MPa; No additional thermal input	5	1.600	3.100	320	620	0.52	10
Rh/CexWO ₃	Light Source: Xe lamp (300-1000 nm) 1.8 W · cm ⁻² ; P = 1 atm; No additional thermal input	50 (no Rh wt.%)	-	-	88.5 mmol h ⁻¹ g _{Rh} ⁻¹	152.3 mmol h ⁻¹ g _{Rh} ⁻¹	0.59	11
Pt/TaN	Light Source: LA-251Xe lamp with L42 + HA30 filters (500–600 nm; 0.42 W cm ⁻²); P = 1 atm; T = 500 °C	100	6.600	7.200	66	72	0.92	12
Rh/SrTiO ₃	Light Source: 150 W Hg–Xe lamp; P = 1 atm; No additional thermal input	5	0.270	0.275	54	55	0.98	13
Ni/(PSC) CeO ₂	Thermocatalysis; P = 1 atm; T = 450 °C	-	-	-	32	48	0.67	14

2c. In Fig. 2d, what is the reason that CW mode (without plasma) showed a higher performance compared to the thermo catalysis?

Reply: We are grateful for the reviewer’s insightful comments and the opportunity to delve deeper into the nuances of laser catalysis.

The observation that continuous wave (CW) laser mode, which lacks plasma generation, outperforms traditional thermocatalysis is indeed intriguing. The key distinction lies in the nature of heat application: CW laser catalysis induces localized heating, concentrating energy within a smaller area, whereas thermocatalysis distributes heat more broadly through electric heating.

Localized heating by the CW laser may enhance the catalytic activity by creating a more favorable environment for the reaction to proceed. This is due to the potential for higher local temperatures and thermal gradients, which can influence the reaction kinetics and pathways. The localized heat could facilitate more efficient energy utilization and possibly activate additional reaction sites that are not as effective under the uniform conditions of thermocatalysis.

As the reviewer stated in question 1f “I stated in the premise, it is possible that the local heating by the laser will increase the activity compared to larger-area heating.” Your hypothesis that local heating can benefit photothermal catalysis is indeed thought-provoking and opens up new avenues for research. It suggests that the spatial distribution of heat, not just the overall temperature, can be a critical factor in catalytic performance. This insight will undoubtedly guide future experiments and could lead to the development of more efficient catalytic systems. We look forward to exploring this concept further and thank you for highlighting its significance.

2d. In Fig. 2d ("Photothermal catalysis"), it is stated that the low activity is due to weak light and low temperature. If so, this comparison is not fair. 300W Xe lamps would emit light at 15 W or more, and the temperature can reach 800°C or more when one uses a suitable optical setup. Why do the authors use such low-intensity conditions? (It would be better to show the measured temperature in the photothermal conditions).

Reply: We would thank for the reviewer's precious comments and suggestions. The photothermal catalysis conditions used in the early stage were indeed suboptimal. According to the reviewer's suggestion, we re-evaluated the performance of photothermal catalytic DRM using a 300 W Xe lamp with a plano-convex lens. Limited by the Xe lamp condition in laboratory, the optical power density of 3 W cm⁻² irradiation was achieved to perform the photothermal catalytic DRM. The photothermal temperature characterized by infrared thermal imaging is shown in **Supplementary Fig. 8**, and a stable photothermal temperature of 361°C was obtained within 20 s via the photothermal conversion of Mo₂C. As the performance results shown in **Fig. 2d**, even when heated to 361 °C under 30 solar light intensities, the activity of Mo₂C in the photothermal catalytic DRM was only 0.251 mmol h⁻¹ g⁻¹ (yield of H₂) and 0.824 mmol h⁻¹ g⁻¹ (yield of CO). This shows that the temperature increase of Mo₂C under ordinary light is limited, and also confirms that the high energy density of pulse laser is the key to generating localized high temperature and plasma.

According to the reviewer's suggestion, we have supplemented the measured temperature in the photothermal conditions in **Supplementary Fig. 8**, and the photothermal catalytic performance in **Fig. 2d** was re-corrected and compared with the photothermal performance data under 30 solar light intensities.

[Minor point].

3 Significant figures should be taken into account. For example, 14300.8 mmol h⁻¹ g⁻¹.

Reply: We appreciate the reviewer's insightful comment and apologize for the inconsistency in significant figures. We have checked thoroughly and modified the production rate data to three decimal places and the yield data to one decimal places.

4 Please check the equations for "total energy efficiency" and "electricity cost". The units seem to be inappropriate.

Reply: We would thank for the reviewer's precious comments. We are sorry for the error of the equations and inconvenience. We have checked and revised the equations on Supplementary Information page 4, which replacing the "n_{CH₄,converted}", "n_{CO₂,converted}" by "r_{CH₄,converted}", "r_{CO₂,converted}" as following:

The total energy efficiency of laser-catalysis and thermocatalysis was calculated using:

$$E = \frac{r_{\text{CH}_4,\text{converted}} + r_{\text{CO}_2,\text{converted}}}{P_{\text{output}}}$$

The total energy efficiency of photocatalysis was calculated using:

$$E = \frac{r_{\text{CH}_4,\text{converted}} + r_{\text{CO}_2,\text{converted}}}{P_L}$$

The electricity cost of laser-catalysis, thermocatalysis, and photocatalysis was calculated using:

$$C = \frac{r_{\text{CH}_4, \text{converted}} + r_{\text{CO}_2, \text{converted}}}{P_E}$$

Δp refers to the percentage change of a product in the reactor; F_{in} and F_{out} refer to the reaction gas flow rate of inlet and outlet, respectively; $r_{\text{CH}_4, \text{converted}}$ and $r_{\text{CO}_2, \text{converted}}$ refer to rate of conversion of CH_4 and CO_2 (mmol s^{-1}), respectively; P_{output} refers to the output power of the pulsed laser (16 W) and the fixed-bed reactor (496 W); P_L refers to the luminous power of the Hg–Xe lamp (60 W, output power of the 150 W Hg–Xe lamp); P_E refers to the electric power of the pulsed laser (1200 W), the fixed-bed reactor (730 W), and the Hg–Xe lamp (150 W).

5 In Fig. 2a and b, the authors describe the horizontal axis as "catalytic area", but I think it is inappropriate. It is not common to think that the irradiated area and the "catalytic area" are the same because the temperature outside the irradiated position also rises.

Reply: We appreciate the reviewer's attention to the terminology used in our manuscript and apologize for any confusion caused by the terms "catalytic area" and "irradiation area." To clarify, we have identified three distinct areas in our experiments:

1. **Tablet Area:** This refers to the actual area of the tablets pressed from $\text{Mo}_2\text{C}/\text{BaSO}_4$, which is represented on the horizontal axis in Fig. 2a and 2b.
2. **Thermal Area:** Illustrated in Fig. 3b-c, this area encompasses the region affected by the temperature changes during the experiment, which may extend beyond the irradiated zone.
3. **Laser irradiation Area:** Shown in Fig. 1i, this is the specific region directly exposed to the laser's energy.

To enhance understanding, we have also included a schematic representation of various catalytic testing modes (Fig. R5 as Supplementary Fig. 4) alongside the corresponding catalytic activities (presented in Fig. R6 as Supplementary Fig. 5 and Tables R1-R3 as Supplementary Tables 1-3). We have revised the manuscript to clearly differentiate these areas and ensure that the experimental details are accurately conveyed. The term "catalytic area" has been adjusted to reflect the tablet area, and additional explanations have been provided to distinguish it from the thermal and irradiated areas. We hope this addresses your concerns and enhances the clarity of our findings.

	Sample 1	Sample 2	Sample 3	Sample 4
Tablet Area (mm ²)	12.57	7.07	3.14	0.79
Thermal Area (mm ²)	0.95	0.95	0.95	0.79
Irradiation Area (mm ²)	0.69	0.69	0.69	0.69

Fig. R5. Schematic representation of various catalytic testing modes. Gray circle represents the catalyst tablet area, Light red circle represents the thermal area of the catalyst tablets, Red circle represents the irradiation area of laser.

Fig. R6. Catalytic activity of the samples using the Mo₂C mass in three areas.

Table R1. Catalytic performance of the samples calculated by total Mo₂C mass within tablet area.

Sample	Tablet Area (mm ²)	Mo ₂ C Mass (mg)	Product rate (mmol/h)		Activity (mmol/h/g)	
			H ₂	CO	H ₂	CO
1	12.57	1.91	7.091	7.611	3712.0	3984.3
2	7.07	1.07	7.330	7.080	6850.5	6616.8

3	3.14	0.48	6.864	7.176	14291.7	14937.5
4	0.79	0.12	4.201	4.802	35000.0	40000.0

Table R2. Catalytic performance of the samples calculated by effective Mo₂C mass within thermal Area.

Sample	Thermal Area (mm ²)	Effective Mo ₂ C Mass (mg) ^a	Product rate (mmol/h)		Activity (mmol/h/g)	
			H ₂	CO	H ₂	CO
1	0.95	0.14	7.091	7.611	50642.9	54357.1
2	0.95	0.14	7.330	7.080	52357.1	50571.4
3	0.95	0.14	6.864	7.176	49000.0	51214.3
4	0.79	0.12	4.201	4.802	35000.0	40000.0

a. Effective Mo₂C Mass is defined as the amount of Mo₂C covered within a thermal area of 0.95 mm².

Table R3. Catalytic performance of the samples calculated by effective Mo₂C mass within irradiation Area.

Sample	Irradiated Area (mm ²)	Effective Mo ₂ C Mass (mg) ^b	Product rate (mmol/h)		Activity (mmol/h/g)	
			H ₂	CO	H ₂	CO
1	0.69	0.10	7.091	7.611	68173.1	73173.1
2	0.69	0.10	7.330	7.080	70480.8	68076.9
3	0.69	0.10	6.864	7.176	65961.5	68942.3
4	0.69	0.10	4.201	4.802	40384.6	46153.9

b. Effective Mo₂C Mass is defined as the amount of Mo₂C covered within a laser-irradiated area of 0.69 mm².

REVIEWERS' COMMENTS

Reviewer #1 (Remarks to the Author):

The authors strengthen the claim by additional information, suggested by the reviewer. Therefore, the paper can be acceptable, I think.

Reviewer #2 (Remarks to the Author):

I have reviewed the revisions and comments by Li et al. with great attention and am impressed by their thorough consideration and response to my comments. They have diligently addressed my concerns by experimentally defining and clarifying them. In particular, they have supplemented additional experiments to determine the catalytic activity, which I was concerned about, and by transparently calculating the catalytic activity from various benchmarks in supplementary figure 5, they provide clarity that can prevent confusion among researchers and offer insights for further improvements based on this study.

The other responses have also been made in accordance with the Reviewer's comments, leading to believe that the current manuscript is suitable for publication in Nature Communications.

REVIEWERS' COMMENTS

In this response letter, the reviewers' comments are presented in *black italics*, our responses are in blue.

REVIEWERS' COMMENTS

Reviewer #1 (Remarks to the Author):

The authors strengthen the claim by additional information, suggested by the reviewer. Therefore, the paper can be acceptable, I think.

Author reply: We sincerely thank the reviewer for your positive feedback on our work. We are grateful for the valuable comments and suggestions, which have significantly improved our paper to an acceptable standard.

Reviewer #2 (Remarks to the Author):

I have reviewed the revisions and comments by Li et al. with great attention and am impressed by their thorough consideration and response to my comments. They have diligently addressed my concerns by experimentally defining and clarifying them. In particular, they have supplemented additional experiments to determine the catalytic activity, which I was concerned about, and by transparently calculating the catalytic activity from various benchmarks in supplementary figure 5, they provide clarity that can prevent confusion among researchers and offer insights for further improvements based on this study.

The other responses have also been made in accordance with the Reviewer's comments, leading to believe that the current manuscript is suitable for publication in Nature Communications.

Author reply: The reviewer's positive feedback on our revised manuscript is greatly appreciated. We are grateful for the reviewer's recognition of our efforts and pleased that our revisions have met with approval. The reviewer's valuable comments and suggestions have undoubtedly improved the quality of our work. We believe these enhancements will make a significant contribution to the scientific community.